# Exploring drought hazard, vulnerability, and related impacts to agriculture in Brandenburg

Fabio Brill[1], Pedro Henrique Lima Alencar[2], Huihui Zhang[1], Friedrich Boeing[3,4], Silke Hüttel[5,6], Tobia Lakes[1,7]

[1] Geography Department, Humboldt-Universität zu Berlin, Berlin, 10099, Germany
[2] Department of Ecohydrology and Landscape Assessment, Technical University Berlin, Berlin, 10623, Germany
[3] Department Computational Hydrosystems, Helmholtz Centre for Environmental Research (UFZ), Leipzig, 04318, Germany
[4] Institute for Environmental Science and Geography, University of Potsdam, Potsdam-Golm, 14476, Germany
[5] Department of Agricultural Economics and Rural Development, University of Göttingen, Göttingen, 37073, Germany
[6] Faculty of Agriculture, University of Bonn, Bonn, 53115, Germany
[7] Integrative Research Institute on Transformations of Human-Environment Systems (IRI THESys), Humboldt-Universität zu Berlin, Berlin, 10099, Germany

Correspondence to: Fabio Brill (fabio.brill@hu-berlin.de)

**Abstract.**

Adaptation to an increasingly dry regional climate requires spatially explicit information about current and future risks. Existing drought risk studies often rely on expert-weighted composite indicators, while empirical evidence on impact-relevant factors is still scarce. The aim of this study is to investigate to what extent hazard and vulnerability indicators can explain observed agricultural drought impacts via data-driven methods. We focus on the German federal state of Brandenburg, 2013-2022, including several consecutive drought years. As impact indicators we use thermal-spectral anomalies (LST/NDVI) on field level, and empirical yield gaps from reported statistics on county level. Empirical associations to the impact indicators on both spatial levels are compared. XGBoost models explain up to about 60% variance in the yield gap data (best $R^2 = 0.62$). Model performance is more stable for the drought years, and when using all crops for training rather than individual crops. Meteorological drought in June and soil quality are selected as strongest impact-relevant factors. Rye is empirically found less vulnerable to drought than wheat, even on poorer soils. LST/NDVI only weakly relates to our empirical yield gaps. We recommend comparing different impact indicators on multiple scales to proceed with the development of empirically grounded risk maps.

# 1 Introduction

Agricultural drought risk mapping is essential for spatial prioritization of adaptation actions and measures, and particularly to raise awareness of stakeholders throughout the social-ecological system (Mishra and Singh, 2011; Blauhut, 2020; Kim et al., 2021). In the light of climate change, droughts are expected to occur in higher frequency and unprecedented magnitudes, which poses a major challenge for risk management (Hanel et al., 2018; Hari et al., 2020; Satoh et al., 2022; Kreibich et al., 2022). Risk in this context can be conceptualized as potential for negative impacts, assembled from the components hazard, exposure, and vulnerability – while definitions of terms have shifted over the years, the recent guideline by the Intergovernmental Panel on Climate Change (IPCC) is very clear on that matter (Reisinger et al., 2020). A sound understanding of hazard thresholds and vulnerability conditions associated with impacts under droughts (hereinafter "impact-relevant factors") is thus urgently needed to provide reliable risk maps and move towards impact-based forecasting (Sutanto et al., 2019). However, many drought risk maps are still being produced by more or less arbitrary weighting of indicators to a composite score (Kim et al., 2015; Dabanli, 2018; Kim et al., 2021; Khoshnazar, 2023), sometimes based on expert opinion (Frischen et al., 2020; Abdullah et al., 2021; Stephan et al., 2023), or by process-based models for individual agricultural crops (Söder et al., 2022). A review of international examples found that drought studies in particular do often neither define their target of investigation in sufficient detail, nor include any sort of validation, thereby making the results difficult to interpret and use (Hagenlocher et al., 2019). Such aggregated indicators could harm more than they help by masking important differences between areas (Jhan et al., 2020). For Brandenburg, our study region, Ihinegbu and Ogunwumi (2022) produced a drought event map based on weighting of the normalized difference vegetation index (NDVI), land surface temperature (LST), and rainfall, without considering vulnerability or impacts. We suggest that drought risk mapping should be more closely related to investigations of actual hazard-impact relationships.

Droughts are natural hazards with a relatively slow-onset character, although there is recently more scientific attention towards flash droughts (Alencar and Paton, 2022). Distinguished are purely meteorological droughts, soil moisture droughts, hydrological low flow in rivers, as well as socio-economic droughts that impose consequences on the broader population and might lead to water conflicts (Wilhite and Glantz, 1985). For agriculture, the direct biophysical drought impacts arguably start once water availability restricts plant growth. Depending on the drought intensity, duration, and timing within the plant phenological stage, crop health is affected, which translates into yield levels, product quality and ultimately prices (Santini et al., 2022). Historically, droughts are associated with famine and high death tolls (Mishra et al., 2019; Contreras, 2019). With modern disaster response, the impacts usually stay on the economic level, but also monetary loss can have severe consequences for individuals, businesses, and entire regions, that are to be anticipated and managed proactively (Erfurt et al., 2019; Krishnamurthy et al., 2022). While there are mechanisms to partially compensate losses due to extreme events (European Commission, 2023), a notable residual business risk remains with the farms – potentially leading to stress and anxiety experienced by farmers (Austin et al., 2018; Abunyewah et al., 2024). Indirect effects are then propagated along the value

chain and within the affected region. More than 100 billion euro have been attributed to drought events between 1986 and 2016 in the European Union (Blauhut et al., 2016), and severe increases of economic impacts are projected for climate change scenarios without adaptation (Naumann et al., 2021). In the German federal state of Brandenburg, our study region, the local government spent 72 million euro of compensations to farmers for drought-related losses in 2018 alone, accounting for about 45% of the actual claims of that year (MLUK, 2019). This, however, was only the beginning of a prolonged multi-year drought (Boeing et al., 2022). As an area that was historically water-rich, Brandenburg now needs to prepare for a dryer future (Kahlenborn et al., 2021; MLUK, 2023), making it an interesting case for an empirical study.

Methods for empirically investigating impact-relevant factors for natural hazards range from simple regression to state-of-the-art algorithms from the field of (explainable) artificial intelligence (AI and XAI, respectively). Investigated impacts include for example damage to buildings from river floods (Merz et al., 2013), debris flows (Jakob et al., 2012), or compound events (Brill et al., 2020), as well as casualties from floods (Tellman et al., 2020) and heat (Şalap-Ayça and Goto, 2023), or the occurrence of wildfires (Kondylatos et al., 2022). There have been similar attempts to uncover impact-relevant factors from text reports of past droughts (Stahl et al., 2016; Blauhut et al., 2016; de Brito et al., 2020; Sodoge et al., 2023; Stephan et al. 2023b), and from yield anomalies for selected crops (Sutanto et al., 2019; Peichl et al., 2021; Tanguy et al., 2023). Despite these recent efforts, empirical evidence on regional impact-relevant factors and non-linearities of actual observed drought impacts is still rather scarce (Bachmair et al., 2016; Sutanto et al., 2019; Peichl et al., 2021; Tanguy et al., 2023). The application of AI methods in particular has led to considerable advances on the side on drought hazard monitoring and forecasting in recent years (Prodhan et al., 2022; Kowalski et al., 2023; Zhang et al., 2024). While these methods are very promising, they do rely on the availability of (big) data covering the processes of interest. On the side of vulnerability and impact-relevant factors, a key bottleneck of such data-driven studies is the availability of impact data.

One potential solution to solve the data availability issue is the use of remote sensing data products, from which indicators of crop health can be derived. While there are various potential indicators for mapping drought impacts on crops, the ratio between LST and NDVI is a particularly well-established observable metric for that purpose (McVicar and Bierwirth, 2001; Karnieli et al., 2010; Crocetti et al., 2020). Mid growing season is generally regarded as the most decisive time of observation (Ghazaryan et al., 2020). Reinermann et al. (2019) used remote-sensing time series from 2000 to 2018 and detected negative vegetation anomalies in Germany during summer months, particularly in the drought year 2018. The correlation strength of drought indicators to yields was found to increase over time (Lüttger and Feike, 2018). However, most data-driven studies using earth observation merely model the occurrence of drought or treat anomalies of spectral indicators as "observed impact" without proper comparison to yields (Houmma et al., 2022).

Based on these identified gaps, the aim of this study is twofold: (1) we investigate the recent drought years in Brandenburg by combining indicators on hazard, vulnerability, and impacts from multiple data sources, and (2) we derive empirical

relationships of hazard and vulnerability indicators to the different impact indicators by data-driven methods. Additionally, an interactive web map was developed to assist on the exploration of the components of regional drought risk. The findings provide new insights on the complexity of the impact-hazard-vulnerability relationship of agricultural droughts for our study region in Brandenburg, as well as on limitations of currently available datasets. This has implications for modelling and monitoring of agricultural drought.

## 2 Material & Methods

### 2.1 Approach & Study Area

To achieve our two objectives, we select a set of indicators based on a literature analysis, including impact indicators on two different levels: field and county level. Spatiotemporal patterns are investigated by visual inspection. We then conduct data-driven analyses to identify hazard and vulnerability indicators empirically associated to the impact indicators on both levels (Fig. 1). These data-driven analyses consist of correlation checks, machine learning regression and model inspection techniques. In addition to this paper, we provide an interactive web-based visualization tool to foster the exploration of data beyond the printed figures.

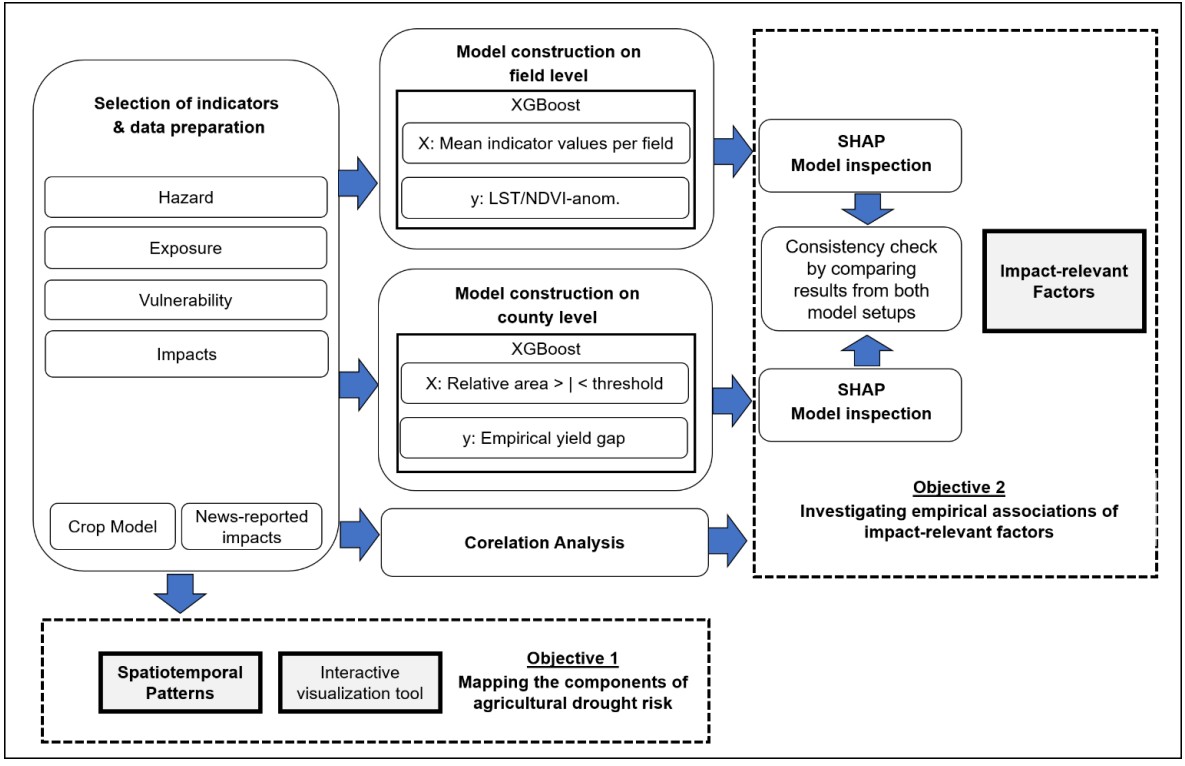

**Figure 1.** Workflow of the presented study

As study region we choose the German federal state of Brandenburg, which has a relevant agricultural sector that has been affected by drought in recent years, and where reported yields as well as high spatial resolution data on grown crops are available. Brandenburg is characterized by flat topography, sandy soils and lakes stemming from the latest ice age, as well as former peatland areas that have been drained over centuries for the purpose of obtaining arable land (LBGR, 2010). The climate is continental and comparably dry for German standards, with averaged precipitation around 600 mm/a, and evapotranspiration around 500 mm/a, including smaller subregions with negative water balances (Germer et al., 2011). Regional climate projections indicate a further reduction in precipitation during the crop growing season, i.e. harsher conditions for agriculture (Kahlenborn et al., 2021; MLUK, 2023). Soil water is generally expected to decrease in the region (Holsten et al., 2009). Agriculture in Brandenburg is primarily rainfed, though, and current priorities of the regional water management suggest that the uptake of large-scale irrigation will not be a realistic option in the near future (MLUK, 2023). Despite this setting, the agricultural sector is very important for the region and its population in the 18 counties (in German: Landkreise, correspond to NUTS-3 regions), with about 1 million ha, one third of the state, used for arable farming (MLUK, 2023). The agricultural sector of Brandenburg has also been identified as highly vulnerable to drought in European- scale studies (de Stefano et al., 2015; Blauhut et al., 2016).

## 2.2 Exposure & Vulnerability Indicators

Spatially explicit information about exposure, i.e. cropped agricultural land, is derived from the Integrated Administration and Control System (IACS), that provides the field-level data on crops for farms which have applied for annual payments within the EU's Common Agricultural Policy (CAP) (Leonhardt et al., 2023). These shapes provide the basis of our field-level analysis. We selected 12 of the most important crop types in Brandenburg in terms of area of production, for which matching information in the yield reports and average values per LBG are available (Table A1). In some cases we only used the winter variety, in other cases we had to merge summer and winter varieties to match the yield reports (Table A2). The 12 crops used in this study are: winter wheat, rye, triticale, oat, winter barley, winter canola, grain maize, sunflower, potatoes, lupines, peas, and sugar beet. The total cropped area covered by our 12 selected crop types is fluctuating in the investigated time period (2013-2022) between about 638.000 to 686.000 ha, with no clear trend. The largest unconsidered fraction is silage maize, which is mostly used as fodder and thus not consistently covered in the reported statistics. Rye is among the most commonly found crops in the region, and regarded as reliable source of income on sandy soils even with little precipitation (LBV, 2024). Wheat is considered to be more demanding but also to realize higher prices. Cultivation of potatoes and sugar beet has been drastically reduced over the last decades, partially owing to the increasingly dry climate (LBV, 2024). Farm level product prices were purchased from the company Agrarmarkt Informations-Gesellschaft (AMI) (cf. LELF, 2021 for publicly available data until 2020).

Vulnerability indicators attempt to capture the relevant characteristics that shape the relationship between hazard intensity and impacts. We compiled a list of environmental and socio-economic indicators and their assumed direction of influence on agricultural drought vulnerability (cf. Walz et al., 2018; Meza et al., 2019; Frischen et al., 2020; Zhou et al., 2022; Stephan et al., 2023). A gridded estimate of agricultural soil quality (in German: Ackerzahl, AZL) is available in 5 m resolution (Schmitz and Müller, 2020). Based on the AZL, 5 different agricultural production areas (in German: Landbaugebiete, LBG), are classified, for which average yields for the most important crops are published (LELF, 2016). As a specific water-related indicator we include the plant-available water capacity (in German: nutzbare Feldkapazität, NFK) (BGR, 2015). To capture potential water accumulation in the landscape, we further derived the topographic wetness index (TWI) from a digital elevation model (BKG, 2017). We extracted mean values of AZL, NFK, and TWI per agricultural field for the available point in time, assuming that they do not change over time. Other indicators, in particular the socio-economic datasets, were only available per county for Brandenburg. This restricted their use to simple correlation analysis with impact indicators on the same spatial level. Large parts of Brandenburg are classified as "disadvantaged area" due to rather poor soils – the exception here being the northeastern counties Uckermark and Märkisch-Oderland. These two counties also exhibit the highest scores for secured succession (along with Potsdam), despite long-known problems with general unemployment in the Uckermark (10.7% in the year 2022). Smaller strips and patches of high quality LBG-1 soils are found in the West (Fig. 2). The spatial distribution of crop types partially reflects these patterns, e.g. winter wheat is typically grown on high-quality soils, making it the dominant crop type in the abovementioned areas, while rye is most common throughout the rest of Brandenburg on poorer, sandy soils.

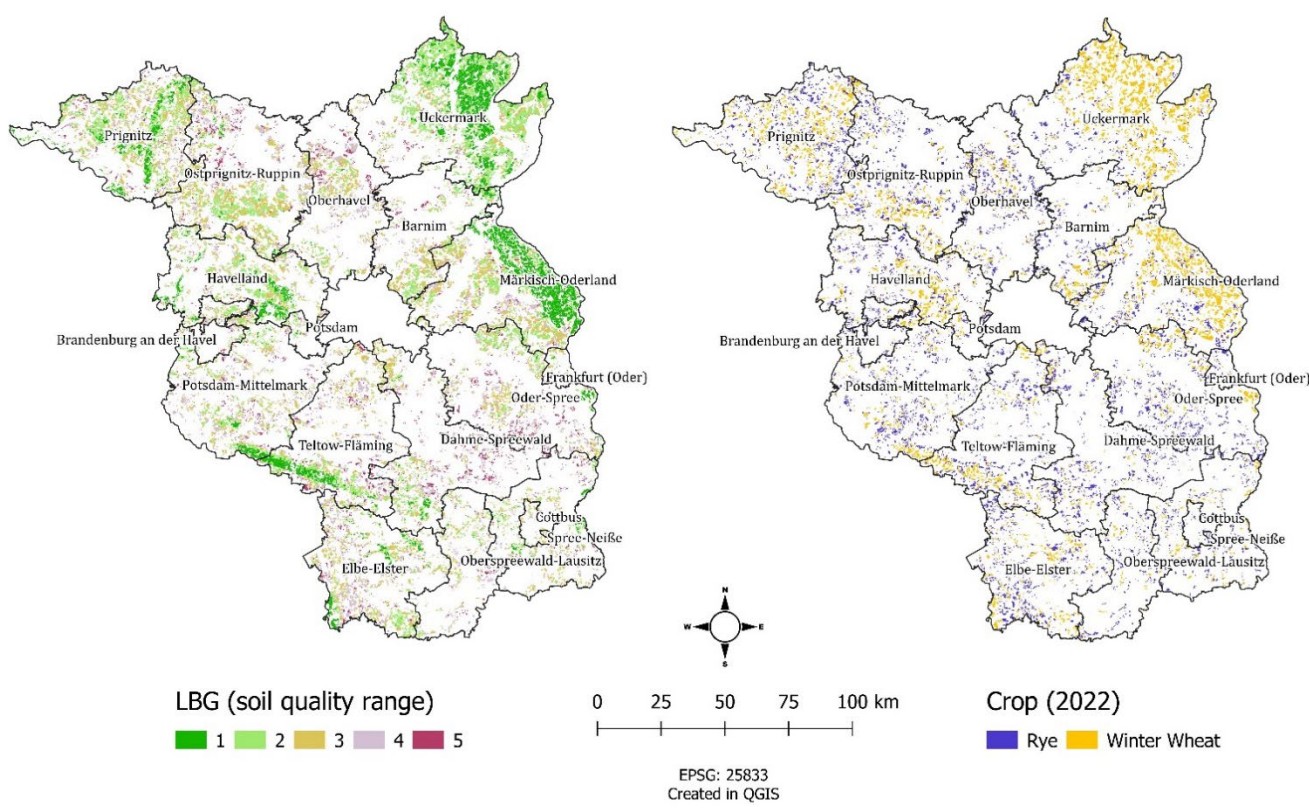

**Figure 2.** Spatial distribution of agricultural soil quality (LBG). Distribution of winter wheat and rye in the year 2022.

## 2.3 Hazard indicators: SPEI and SMI

The Standardized Precipitation Evaporation Index (SPEI) captures both precipitation and potential evaporation and has evolved as one of the most commonly used meteorological drought indicators in recent years (Vicente-Serrano et al., 2010; Rossi et al., 2023; Tanguy et al., 2023). Monthly values of SPEI-1 (one-month accumulation SPEI) used in this study are at 10 km grid resolution from 2013 to 2022, based on the E-OBS dataset (Cornes et al., 2018). The calculation details are described in Zhang et al. (2024). As the harvest of main crops in the region typically starts in July, we used data for the months March to July. Negative SPEI values indicate meteorological water deficit. In addition to the monthly SPEI, a metric of growing season drought magnitude was computed as the sum of SPEI-1 < -0.5 over the period between March and July (SPEI-Magnitude) (cf. Wang et al., 2021 for SPEI thresholds).

Regarding soil moisture droughts, the model-based German drought monitor developed at the Helmholtz Centre for Environmental Research (UFZ) is the most established regional product (Samaniego et al., 2013; Zink et al., 2016; Boeing et al., 2022) and has already been used for similar purpose (Peichl et al., 2021). Identical to the SPEI data, we use monthly values and a growing season aggregation of drought intensity derived from the soil moisture index (SMI) for the top soil (25 cm),

again from March to July (Eq. 1). To add some information on slower long-term drought processes (i.e. accumulation and lag time), we further include the annual drought magnitude for the total soil (up to 1.8 m depth), which is temporally aggregated from April to October (SMI-Total).

$$SMI = \frac{1}{d \cdot A} \sum_{t_0}^{t_1} \int_A \left[ \tau - SMI_i^*(t) \right]_+ \tag{1}$$

where $\tau$ is the drought threshold, $SMI^*$ is the raw soil moisture index, and $d$ and $A$ refer to the duration and area of potential aggregation, respectively. A value of 0 for all SMI-based features thus means, that none of the values were below drought threshold $\tau$. We use $\tau = 0.2$ (20th percentile), which is a common value for drought analysis adopted in the literature (e.g. US drought monitor, Svoboda et al, 2002). For more details, the interested reader is referred to Boeing et al. (2022).

## 2.4 Impact indicators: crop health observations and empirical yield gaps

### 2.4.1 LST/NDVI Anomaly

As an indicator of crop health, the ratio of LST and NDVI between May and June, i.e. roughly mid growing season, of each year (2013-2022) was obtained from Landsat-8 satellite imagery, using all images of the T1_L2 collection. This dataset already includes processed LST (Cook et al., 2014). Pre-processing and cloud-masking were conducted within the Google Earth Engine (Gorelik et al., 2017). The temporal aggregation of the satellite data is necessarily a compromise: a comparison between years gets more precise when the interval is shorter, but to smooth out potential variations in overpass and cloud cover, as well as disturbances on individual pixels, mean values across several weeks are generally more trusted (Ghazaryan et al., 2020). Images were downloaded in 30 m spatial resolution and then aggregated on individual fields. A small fraction of fields had to be discarded due to missing data, e.g. because of cloud cover, and we continued the statistical analysis with the remaining ones. As different crop types exhibit characteristic spectra, we further computed the anomalies of LST/NDVI over the entire observation period stratified by crop type (Eq. 2). By doing so, the resulting anomalies (LST/NDVI-anom.) are comparable among different crops.

$$LST/NDVI_{anom,f,y} = \frac{LST/NDVI_{c,f,y} - \overline{LST/NDVI}_c}{\overline{LST/NDVI}_c} \tag{2}$$

where $\overline{LST/NDVI}_c$ is the is the area-weighted mean for a given crop across all years, and the subscripts c, f, and y denote crop, field, and year, respectively.

### 2.4.2 Empirical yield gaps

We further calculated empirical yield gaps per county for 12 crops for the last 10 years (2013-2022), by subtracting actual reported yields (total production in tonnes) by the regional statistical authority (Amt für Statistik Berlin-Brandenburg, 2022; Alencar, 2022) from an estimate of expected yields under non-drought conditions. We refer to expected yields as the product of cropped area (per crop type in a given year) and the respective 5-year average yield (tonnes per hectare) per LBG from the

210 time 2010-2014. The expected yields are computed on field level and then aggregated on the level of counties, to be comparable to the reported yield data. The empirical yield gap divided by the expected yield we call "relative gap". A relative gap value of 1 thus implies that all yield was lost, while a value of -1 implies that double the expected amount was reported, and a value of 0 indicates a perfect match between expected and reported numbers. To correct for differences in the total area reported in IACS as compared to the yield reports, we added the difference in area per crop, multiplied with the average yield per hectare

of that crop within the respective county as derived from the data. Some minor assumptions had to be made to merge crop types reported in IACS with the reports like neglecting spelt in the statistics for wheat (details in Appendix A). Multiplication of the empirical yield gaps and prices of the respective year results in a total estimate of monetary loss in euro. As not all of the 12 considered crops are grown in all regions in every year, the total monetary loss estimate can be based on partially different crops per region. We assume this reflects the real agroeconomic situation in each region.

### 2.4.3 Comparison to external data

For a plausibility check, we compared the resulting empirical yield gaps and loss estimates to regional newspaper reports. For individual crops (rye, wheat, maize, barley) we were able to additionally calculate the potential production (PP) and water-limited production (WLP) by the process model WOFOST on a 2 km grid resolution (Jänicke et al., 2017; de Wit et al., 2019).

If our expected yields from the pre-drought years are realistic, they should be similar to the potential production. Crop growth in WOFOST is modelled from irradiation, temperature, $CO_2$ concentration, plant characteristics, seeding date, and availability of water. The physically modelled potential production from this simulation matches very well with the expected yields derived by our empirical approach for soil quality range LBG-2 in the case of wheat and barley, and LGB-1 in the case of rye (Fig. 3). We are thus confident that our approach produces estimates in a realistic range. Only for maize the modelled potential

production is higher than the average values for Brandenburg suggest on any soil type. This comparison also underlines that it is important to account for the soil quality range, and thus our empirical approach appears more realistic than this particular WOFOST simulation. For further comparison we use the newspaper reported impact score by Sodoge et al. (2023), for the category "agriculture". All data used is summarised in Table 1.

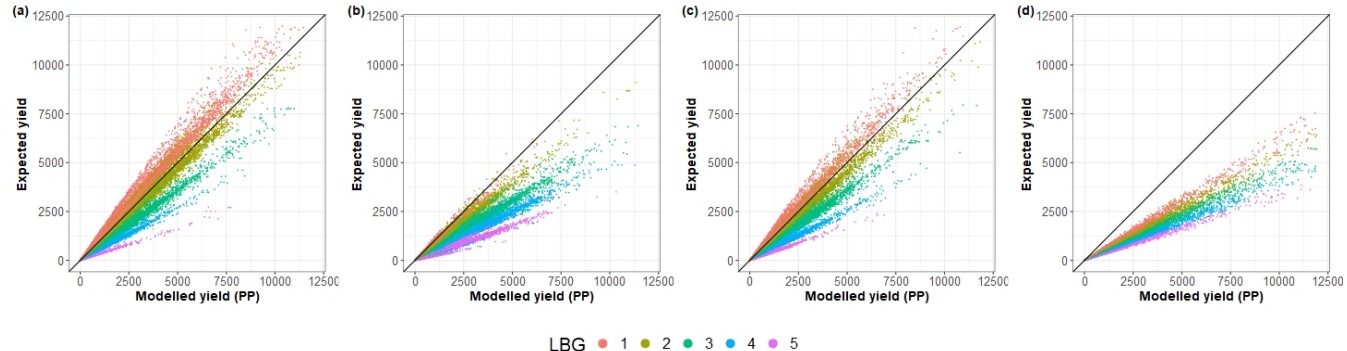

**Figure 3.** Comparison on field level (a) wheat (b) rye (c) barley (d) maize. The original resolution of the crop model is 2 km

**Table 1**. Indicators and data sources

| Category | Abbreviation | Indicator Description | Spatial Res. | Data source and references |
|---|---|---|---|---|
| Hazard | SPEI (monthly) SPEI Magnitude | Standardized Precipitation-Evaporation Index, Sum of SPEI < -0.5, March–July | 10 km | Cornes et al. (2018), Zhang et al. (2024) |
| | SMI (monthly) SMI Magnitude SMI Total | Soil Moisture Index, top soil (25 cm) Top soil, March–July Total soil (max. 1.8 m), April–October | 4 km | Boeing et al. (2022), UFZ Drought Monitor / Helmholtz Centre for Environmental Research https://www.ufz.de/index.php?en=37937 |
| Exposure | - | Agricultural land, on which one of the 12 selected crops is reported in the IACS dataset | Fields (vector) | Integrated Administration and Control System (IACS) MLUK (2022c), Leonhardt et al. (2023) |
| Impact | LST/NDVI LST/NDVI-anom. | Land Surface Temperature / Normalized Difference Vegetation Index. Mean of May–June Anomalies per crop | 30 m | Landsat-8, collection: Landsat/LC08/C02/T1_L2 Courtesy of the U.S. Geological Survey (USGS) accessed via Google Earth Engine https://developers.google.com/earth-engine/datasets/catalog/LANDSAT_LC08_C02_T1_L2#description |
| | Empirical yield gap Relative yield gap | Expected - Reported (Expected - Reported) / Expected where Expected is based on 5-year average hectare yields per LBG and the annual area in ha | County | 5-year average hectare yield per LBG: LELF (2016) Reported: Amt für Statistik Berlin-Brandenburg (2022) Compiled by Alencar (2022) https://github.com/pedroalencar1/CropYield_BBr |
| | Loss estimate | Sum (empirical yield gap * farm level price), for all crops reported in a county per year | County | Farm level prices: AMI, cf. LELF (2021) for publicly available data until 2020 |
| | PP WLP Modelled Gap | Potential production from a crop model Water limited production PP-WLP | 2 km | WOFOST: de Wit et al. (2019) Forcing: Jänicke et al., (2017) |
| | Newspaper reported- impacts | Number of newspaper articles reporting agricultural drought impacts (text-mining based) | County | Sodoge et al. (2023) |
| Environmental Vulnerability | AZL LBG | Agricultural soil quality ("Ackerzahl"), 5-class ordinal range ("Landbaugebiet") | 5 m | Schmitz & Müller (2020) LELF (2021) |
| | TWI | Topographic wetness index | 200 m | BKG (2017) |
| | NFK | Plant available water ("nutzbare Feldkapazität") | 250 m | BGR (2015a) |
| | - | Soil depth | County | BGR (2015b) |
| | - | Soil water erosion | County | BGR (2014a) |
| | - | Soil wind erosion | County | BGR (2014b) |
| | - | Water exchange frequency | County | BGR (2015c) |
| | - | Forest ratio | County | Statistische Ämter des Bundes und der Länder (2020a) |
| | - | Farmland ratio | County | Statistische Ämter des Bundes und der Länder (2020a) |
| | - | Protected area | County | LfU (2020) |
| | - | Disadvantaged area | County | MLUK (2022b) |
| | - | Livestock health | County | Statistische Ämter des Bundes und der Länder (2020b) |
| Socio-economic Vulnerability | - | Secured succession | County | Statistische Ämter des Bundes und der Länder (2020b) |
| | - | Poverty | County | Amt für Statistik Berlin Brandenburg (2019b) |
| | - | Education | County | Statistische Ämter des Bundes und der Länder (2021) |
| | - | Unemployment | County | Statistische Ämter des Bundes und der Länder (2022) |
| | - | Social dependency | County | Eurostat (2021) |
| | - | Agricultural population density | County | Eurostat (2021), Statistische Ämter des Bundes und der Länder (2010) |
| | - | GDP per farmer | County | Eurostat (2022), Statistische Ämter des Bundes und der Länder (2010) |
| | - | GDP per capita | County | Eurostat (2021, 2022) |
| | - | Agricultural dependency for livelihood | County | Statistische Ämter des Bundes und der Länder (2020d) |
| | - | Public participation (voting) | County | Amt für Statistik Berlin- Brandenburg (2019a) |
| | - | Investments in DRR (less favoured areas) | County | MLUK (2022a) |

## 2.5 Statistical procedures and algorithms

Exploratory analysis is conducted by calculating descriptive statistics, correlation matrices and by visual inspection of spatiotemporal patterns in the data. For the latter, we additionally provide a simple web app in R-Shiny. Changes over the investigated years are analysed by plotting the shift of statistical distributions and the temporal evolution of regional mean values. To investigate empirical relations between our hazard and vulnerability indicators and the to our impact indicators, we apply the statistical learning algorithm extreme gradient boosting (XGBoost) (Chen and Guestrin, 2016) combined with the model inspection technique Shapley additive explanations (SHAP) (Shapley, 1953; Lundberg and Lee, 2017). This combination is widely used in the field of XAI and has recently been successfully applied in many different scientific studies to derive insights from complex non-linear and interacting datasets (Yang et al., 2021; Jena et al., 2023; Raihan et al., 2023; Li et al., 2024). XGBoost is an ensemble method based on boosting, i.e. consecutive models are trained on the residuals of the predecessor, thereby increasing the fit step-by-step (as opposed to bagging like in Random Forest, where an ensemble is trained in parallel fashion and aggregated via majority voting). This iterative analysis of errors and weight-adjustment supposedly leads to models that reflect actual patterns in the overall data, rather than random patterns observed in random bootstrap subsets. We use a common tree-based model variant to allow for a hierarchical structure. As sampling scheme, we implemented a nested cross-validation, with an inner loop for hyperparameter optimization and an outer loop to assess the skill on independent holdout sets (not used in parametrization). SHAP values were computed for the best model of each nested iteration, selected by the highest $R^2$ score on the holdout set. The SHAP values represent a game-theoretic estimate of effect size, where the feature values are treated as players that can join a coalition game (model). The resulting values give the expected marginal contribution for each feature value across all possible coalitions, in the unit of the model target, and fulfill the efficiency property, meaning that they sum up to the difference between the overall expected value and the specific model prediction for a set of feature values. By computing these SHAP values for all samples used to construct a model, it is possible to visualize the effect each feature has within the inspected model. Note that this does not necessarily imply insights into processes in nature, but rather into empirical relations in the data as learned by the specific model.

In total our dataset contains 437.476 agricultural fields across 18 counties. With 12 crop types and 10 years the theoretical maximum number of data points on county level is 2160, of which missing entries have to be removed (not all crops grown in all counties in all years). Predictive features on field level are the indicator values. Some feature engineering is necessary to convert the field-level data into features on county level. It is reasonable to assume that damaging processes are more dependent on extreme conditions than on the mean value over a large area. To retain as much information about the hazard distributions, we computed the relative affected area (non-)exceeding specified thresholds in regular intervals (Appendix B). This manual feature engineering resulted in a total of 68 features on county level.

## 3 Results & Discussion

### 3.1 Spatiotemporal patterns of hazard, vulnerability and impact indicators

#### 3.1.1 Temporal evolution on country level

The temporal evolution of mean indicator values for entire Brandenburg suggests that the investigated decade can be divided into a pre-drought phase (2013-2017) and a drought phase (2018-2022) (Fig. 4). In 2013 and 2014 the SMI-Total is close to 0, observed vegetation health is at its maximum (i.e. negative LST/NDVI-anom.), essentially no impact-related statements are captured in the newspaper text-mining data, and our economic calculation even estimates a plus of about 100 million euro compared to the expectations. Especially 2014 indeed made headlines with record-breaking (positive) yields (Agrarheute,

2014). However, the crop model WOFOST still estimates a gap between potential production and water-limited production in that year, and also in the SPEI magnitude there is some drought signal visible. We interpret this as locally and temporally constrained meteorological effects that did not propagate to the soil and consequently did not have a negative effect on crop health and yields. The year 2017 was then rather wet, which is reflected in SPEI, LST/NDVI, and the media impact statements. However, the soil drought did only decline slightly according to the SMI. From 2018 a multi-year drought started. There seems

to be a temporal lag of 1 year between meteorological and soil moisture drought indicators, likely reflecting the propagation from atmospheric conditions to the deeper soil layers. This is also visible in data for the year 2021, where SPEI-Magnitude indicates a good meteorological water balance, but soil moisture drought stayed. Interesting to note though is that the satellite observations of crop health peak in the same year as the SMI-Total, 2019, while the estimated economic loss (12 crops), as well as the crop model (for wheat) and newspaper reported impacts exhibit peaks at the meteorological drought maximum in

2018. The distribution of LST/NDVI-Anomalies has been shifting towards higher values in recent years – not only the median, but also the upper tail of the distribution became heavier (Fig. 5). This upper part of the distribution is where we expect impacts like reduced yields. The most notable exposure changes over the decade are decreasing trends for rye (-30%), triticale (-22%), winter canola (-26%), sugar beet (-33%), and lupines (-38%), increase of winter wheat (+19%), winter barley (+28%), oat (+44%), peas (+132%), and sunflower (+145%) (Fig. 6). Changes in crop choice may partially reflect a response to experienced

crop-damaging conditions, but are also driven by unconsidered factors such as fertilizer or market prices (e.g. Albers et al., 2017). Our total loss estimate from the 12 crops for Brandenburg 2018 is 132 million euro, which comes close to the official numbers: 72 million euro of compensations have been issued by the state, and this sum was considered to account for about 45% of actual claims (which would translate to a loss of 160 million euro when taken at face value) (MLUK 2019).

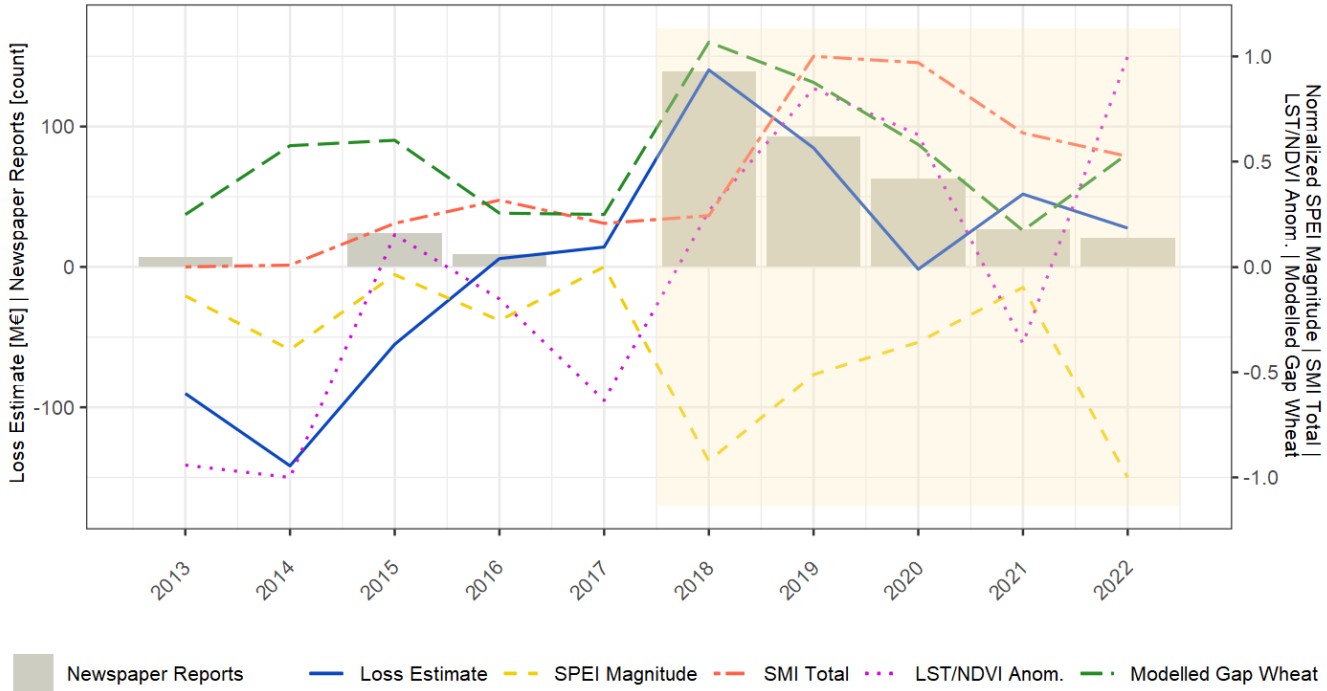

**Figure 4.** Evolution of indicator values over entire Brandenburg. Our loss estimate is given in the original values on the left axis. Bars indicate the number of agricultural impact statements in newspapers on the original scale (left axis). All other indicators were extracted on agricultural fields, area weighted, and scaled to fit the same axis. SPEI-Magnitude has only negative values and is thus scaled to [-1, 0]. SMI-Total and crop model-based gap (PP-WLP) have only positive values and are thus scaled to [0, 1]. LST/NDVI-Anom. has positive and negative values and is thus scaled to [-1, 1].

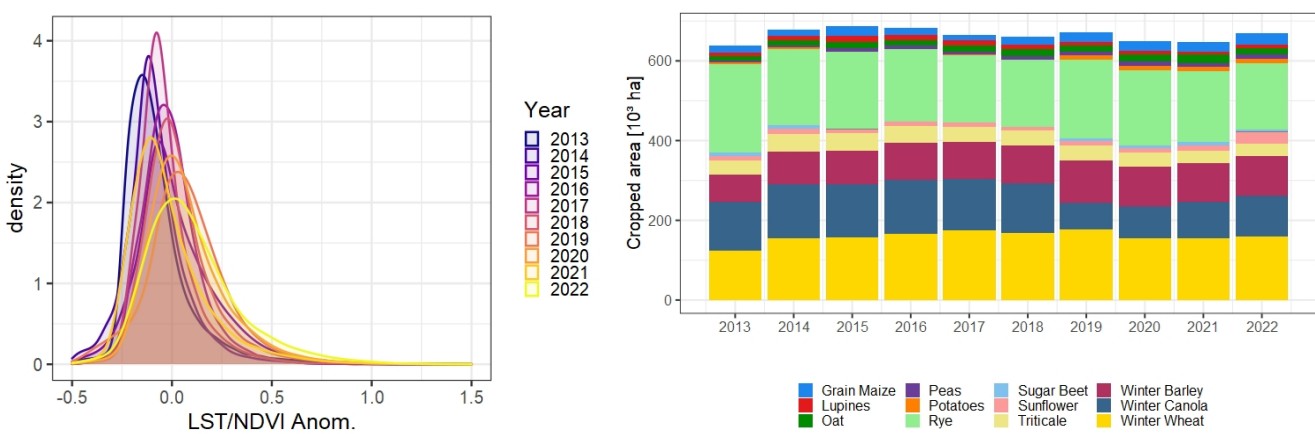

**Figure 5.** Shifting distribution of LST/NDVI anomalies by year. **Figure 6.** Area covered by 12 selected crop types in Brandenburg

Our empirical yield gaps peak in the year 2018 for most crops in most regions, but the variability between counties is high for most crops (Fig. 7). Only winter wheat, winter canola, and winter barley exhibit low to moderate variability between counties. Sugar beet is only reported in a few cases. Plausibility checks against newspaper articles suggest that our relative gap estimates are in a reasonable range: yield reduction for individual crops from 25% to more than 50% have been reported in 2018 and 2019, with winter canola performing worse in 2019 (Agrarheute, 2018; DLF 2019). Grain crops did better in 2022 than 2021, but maize much worse (Tagesschau, 2022). The year 2014 on the other hand is remembered for record-breaking yields with "+24% compared to the previous 5-year average and 11% higher than the previous year", indicating that 2013 was still well above average (Agrarheute, 2014), which is captured in our estimates.

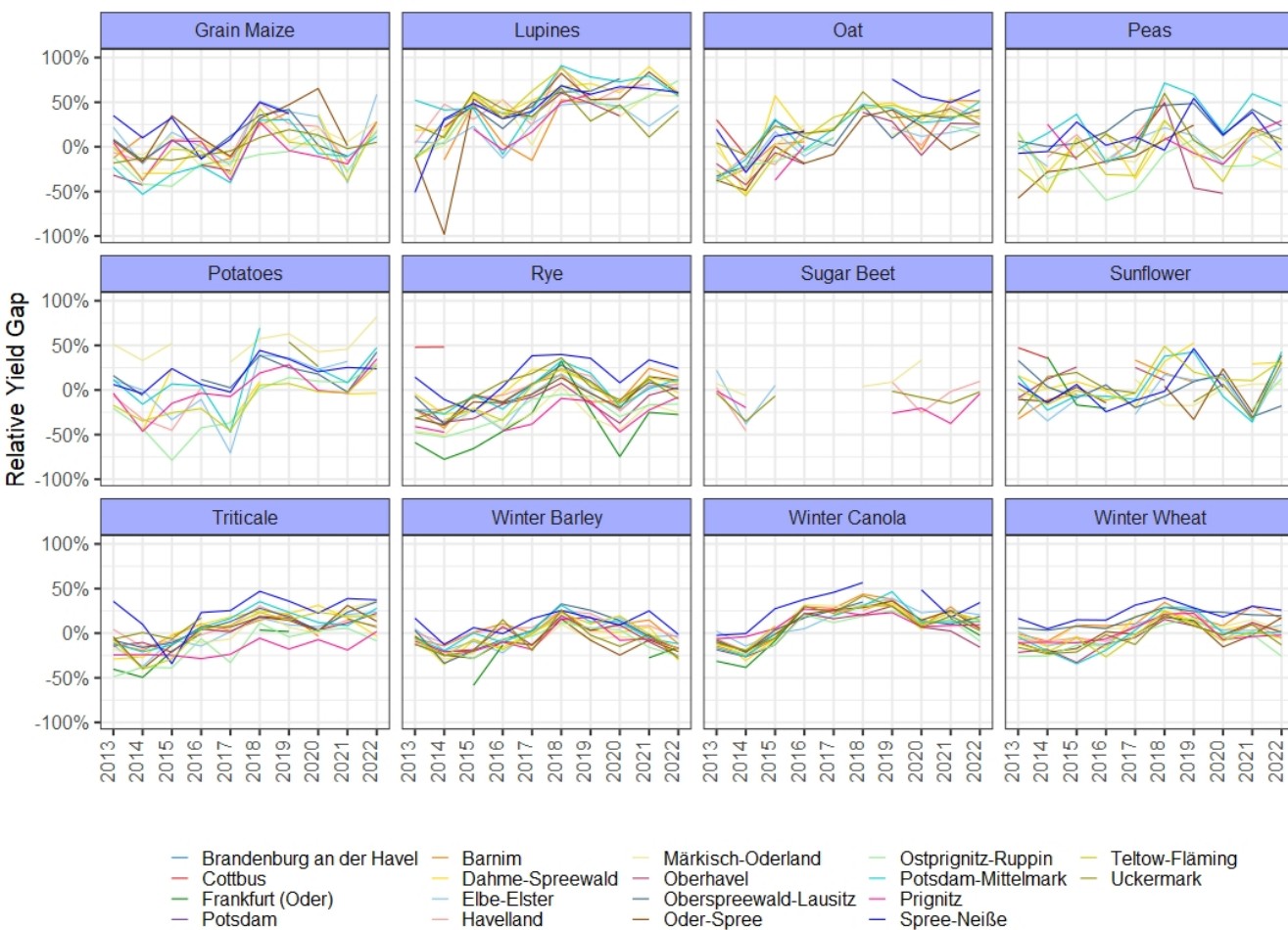

**Figure 7.** Relative yield gaps per county in percent for the 12 investigated crops.

### 3.1.2 Spatiotemporal Patterns

A more facetted picture appears when comparing the spatial distributions of hazard and impact indicators alongside each other for consecutive years (Fig. 8). Essentially the entire state of Brandenburg was affected by meteorological drought in 2018, with the SPEI-Magnitude minimum registered in the South-West. Soils in the South were already dry by then, but severe soil moisture drought throughout the country developed a year later. Contrarily, during the rather rainy year 2021 the accumulated soil drought persisted. When another intense meteorological drought struck in 2022, only the soils in the North had moderately recovered. Annual distributions of LST/NDVI-anom. exhibit small scale variability that is difficult to align with the aggregated hazard indicators. Patches of high anomalies (i.e. supposedly damaged fields) are found scattered across the country, while low anomalies (i.e. supposedly healthy crops) appear to dominate in the areas of good soil quality (cf. Fig. 2). The highest economic loss per hectare is mapped in the southern areas Spree-Neiße and Oberspreewald-Lausitz (the highest absolute loss in the Uckermark, due to the large fraction of agricultural land). While the exceptional years 2018 and 2019 also caused severe losses in the North and West of Brandenburg, the South-East ranks high in the relative loss estimates throughout all of the investigated years. Loss per hectare from our empirical approach is higher than the crop model estimates by Söder et al. (2022), who report separate numbers of around 90 euro per hectare from summer drought plus 60 euro per hectare from spring drought in 2018 in the region. Our estimates refer to the sum of all damaging processes. The socioeconomic vulnerability indicators and low resolution maps for all investigated years can be viewed at https://fabiobrill.shinyapps.io/agrdrought-explorer-brandenburg/, while the high resolution data can be obtained from the GitHub repository https://github.com/fabiobrill/brandenburg-drought-study.

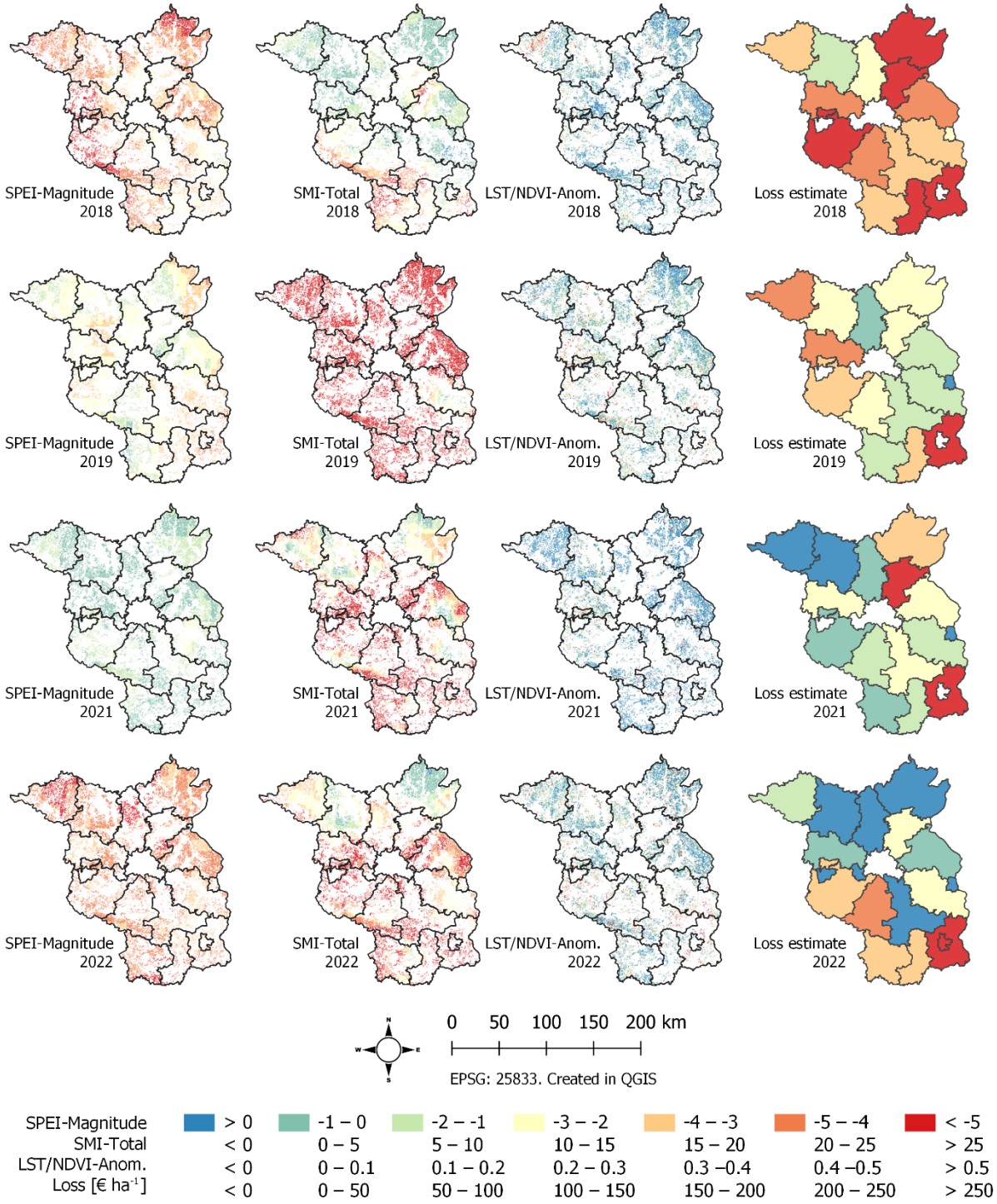

**Figure 8.** Spatiotemporal patterns of aggregated meteorological and soil moisture drought hazard indicators, crop health anomalies, and county-scale loss estimates per hectare.

## 350  3.2 Empirical investigation of impact-relevant factors

### 3.2.1 Relations between indicators on field level

The spatiotemporal patterns suggest non-trivial and multi-way interactive relationships between our chosen hazard, vulnerability and impact indicators. This is further supported by a correlation analysis, which shows that the bivariate linear relations in the data are mostly weak (Fig. 9). Correlations slightly increase when subdividing the data by crop, presumably

because the relationships are more linear for individual crops, however the effect is almost negligible (not shown). The meteorological and soil moisture hazard indicators SPEI and SMI are correlated among each other. Monthly SPEI and SMI are essentially uncorrelated to LST/NDVI-anom. in March, very weakly correlated in April, and moderately correlated in May and June. As the LST/NDVI measurements are also from May and June, the additional correlation in July has to be a spurious effect stemming from the collinearity in the SPEI layers (almost 0.5 between June and July). Raw NDVI – and therefore also

LST/NDVI – is clearly related to AZL, meaning that crops grown on better soils tend to be "greener", with or without drought. This effect is reduced in the anomalies. TWI and NFK exhibit no relation except to AZL. The modelled water-limited production from WOFOST only weakly relates to LST/NDVI (not shown).

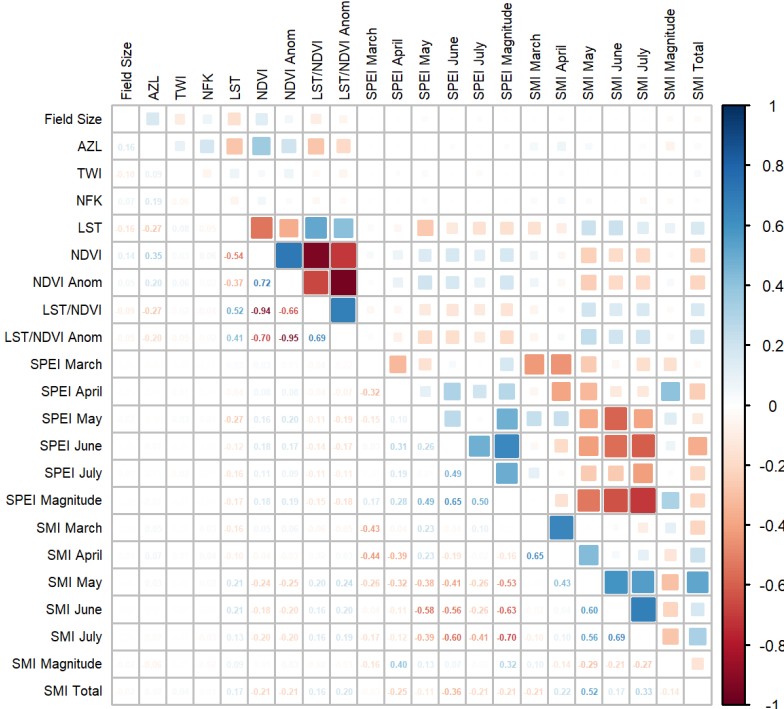

**Figure 9.** Pearson's correlation coefficient on field level data. Almost all correlations are statistically significant due to the high number of samples (n=437.245 complete observations, 474.966 in total).

An XGBoost model trained to predict LST/NDVI-anom. from monthly hazard indicators SPEI and SMI, aggregated SMI-Total, environmental vulnerability factors AZL, TWI, and NFK, as well as crop type, obtains $R^2$ scores around 0.5 (Appendix C). AZL is chosen by the models as most important feature, followed by the categorical variable crop type, while the other environmental vulnerability factors, TWI and NFK, have little influence (Fig. 10). Each dot in these plots corresponds to a sample, and the SHAP values represent the feature effects (conditional expectation) on the predicted quantity, i.e. LST/NDVI-anom. in this case. Interaction plots for crop types highlight that wheat, canola, and barely are grown on relatively good soils, lupines on bad soils, and rye on both (Fig. 11). While the absolute effect of AZL on the predictions is higher than the effect of crop type, particularly wheat is modelled to be impacted more likely than other crops despite growing on better soils (higher AZL).

Some more process understanding about droughts might be distilled from the SHAP dependence plots (Fig. 12). A sharp increase of SHAP values is observed for AZL below 35, meaning that vulnerability is higher on soils below that quality. There is a strong interaction between AZL and SMI-Total, which on its own shows a weakly S-shaped relationship to the LST/NDVI anomaly. A more or less linear response is uncovered for SMI in May, with an offset at 0, i.e. good vegetation health for no drought in May. Meteorological drought in June seems to have a decisive effect in the model, judged by a sharp increase of SHAP values for SPEI < -1. SPEI in March appears to have a damaging effect under too wet conditions (SPEI > +1), which is in line with previous findings by Peichl et al. (2021).

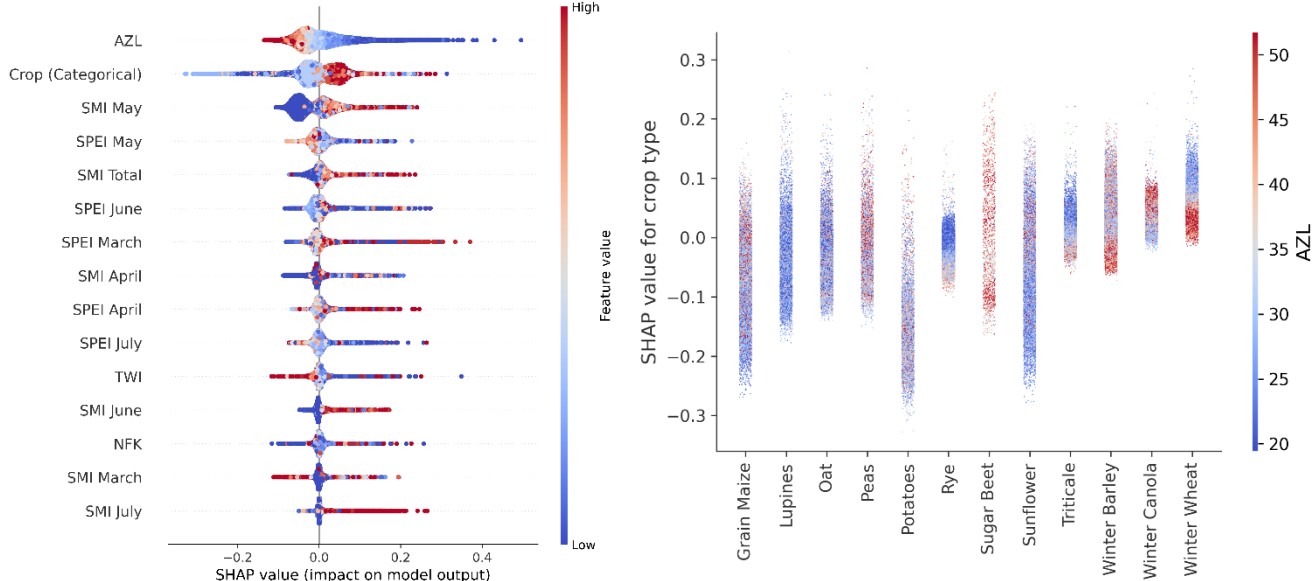

**Figure 10.** SHAP summary plot.          **Figure 11**. Interaction plot for crop type and AZL.

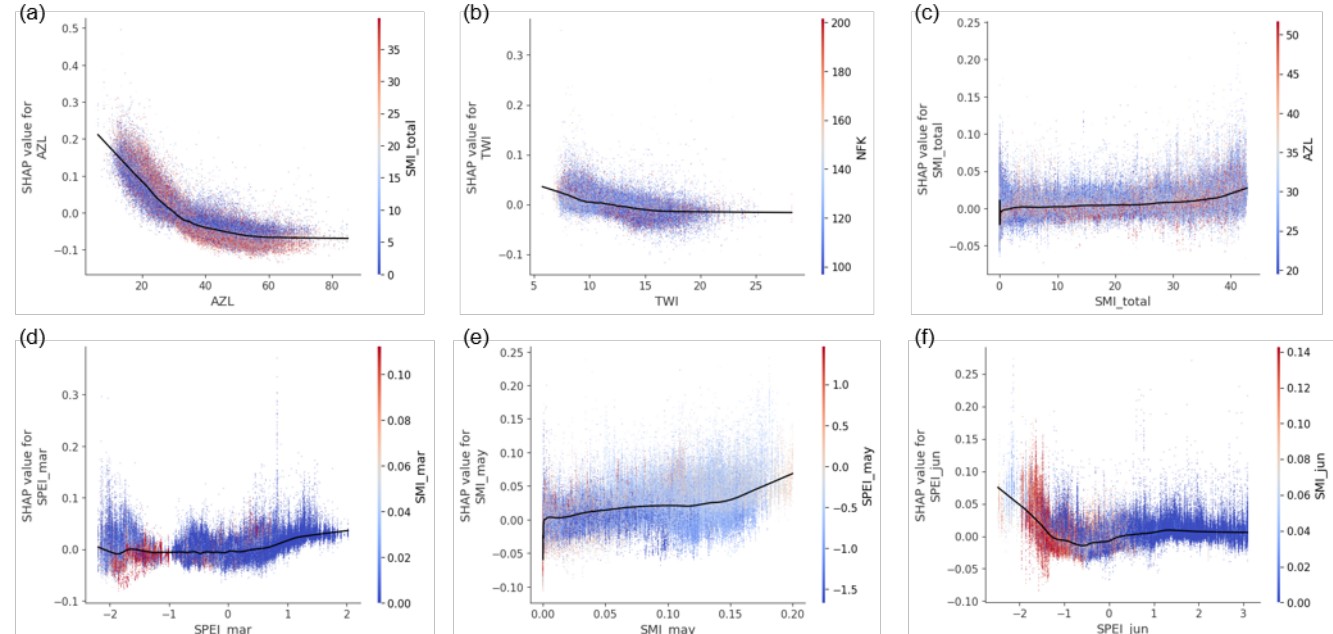

**Figure 12.** SHAP dependence plots for selected features: (a) AZL, (b) TWI (c) SMI Total, (d) SPEI March, (e) SMI May, (f) SPEI June. Centre line derived by a loess regression on the SHAP values. Colour visualizes interaction with a second feature.

From a methodological point of view, it is worth to mention that SHAP plots based on the full dataset exhibit far larger variance on the y-axis than preliminary experiments with only 10% of the data. One reason for this might be the spatial resolution of the features, but we assume that it is also related to the complexity of the regression task. While there are some clear effects in the centre lines, it also becomes obvious that no single feature explains the full data. Several steps in our analysis include simplifications, e.g. calculations using mean values per field imply that an entire field is treated as a unit. For larger fields it might be realistic that only parts are affected, however such effects are below the credible resolution of input data. We acknowledge that particularly for maize, which is typically harvested from September on, a longer observation window might be better suited. Adjusting the remote sensing data to the actual sowing and harvest dates of each crop might improve the results – however, doing so would further complicate the data pre-processing and was considered out of scope of this study. Although agriculture in Brandenburg is predominantly rainfed, a future study could also benefit from spatially explicit information on irrigated areas (Ghazaryan et al. 2022).

### 3.2.2 Relations between indicators on county level

When arranging the 12 crops by correlation among the relative gaps (i.e. each sample referring to a county in a given year), it appears that almost all crops are positively correlated over time, while spatially (and thus spatiotemporally) several groups emerge (not shown). Correlation between the newspaper-based "agriculture" impact score by Sodoge et al. (2023) and our relative economic impact measure (in euro per hectare) over all 12 crops for the years 2013-2022 is 0.75 for entire Brandenburg and 0.53 on county level. When compressing the data to mean values over the entire timespan to merge them with the socio-

economic vulnerability indicators, the highest correlation of newspapers reported impacts is to participation in local politics (0.69). From our data and analysis, we see no meaningful correlation between the vulnerability indicators to reported impacts or calculated losses. A major drawback is the resolution of the indicators. For these reasons they were not included in the following XGBoost regression analysis.


The statistical learning models trained to predict the empirical yield gaps on county level obtain $R^2$ scores around 0.6 when using all features and all data (Fig. 13). Models using only LST/NDVI features as predictors perform poorly ($R^2 \sim 0.2$). It is quite remarkable that a field-level (i.e. high spatial resolution) observation of crop health does not provide more useful information for predicting yield. Models using hazard indicators as predictors perform better. Monthly values of SPEI are

clearly to be preferred over seasonally aggregated magnitude, and the same is true for SMI. However, we observe that models using only SPEI perform slightly better than those using only SMI. One potential reason for this might be that the SMI is itself model-based, which introduces further uncertainty. We find a minor improvement when using both SPEI and SMI, where SMI-total is more relevant than the monthly top soil layers (as complementary information to SPEI). The additional improvement when adding LST/NDVI features on top is almost negligible. Our predictive features explain much more variance

for the drought years 2018-2022 than for the pre-drought years 2013-2017, as expected. Models trained on the full dataset exhibit both higher skill and less variance. A similar effect is observed when training separate models for the different crop types: individual models for winter wheat perform better than individual models for rye, but a lumped model using all crops is much more stable. We explain this by the higher number of training samples in combination with a tree-based model structure that exploits similarities between crops. The $R^2$ skill score of the final model used for inspection via SHAP plots is 0.62 on the

holdout set, i.e. about 60% of the variance in the empirical yield gaps can be explained by our drought-related features, while about 40% remain unexplained. Agricultural crops are highly managed and face numerous threats, not only droughts. It would be unreasonable to assume that drought indicators alone could fully explain real observed yield data. In a similar published attempt, Peichl et al. (2021) report that their best model for winter wheat obtained an $R^2$ of 0.68, which is very close to our best models – however, they do not report any details on the variability of this score. Empirical damage models, such as used

for floods, typically report rather weak model fits (e.g. Wagenaar et al., 2017; Sieg et al., 2017). In the European Drought Risk Atlas, Rossi et al. (2023) do not even report model fit at all, but still uncover plausible impact-relevant factors for droughts.

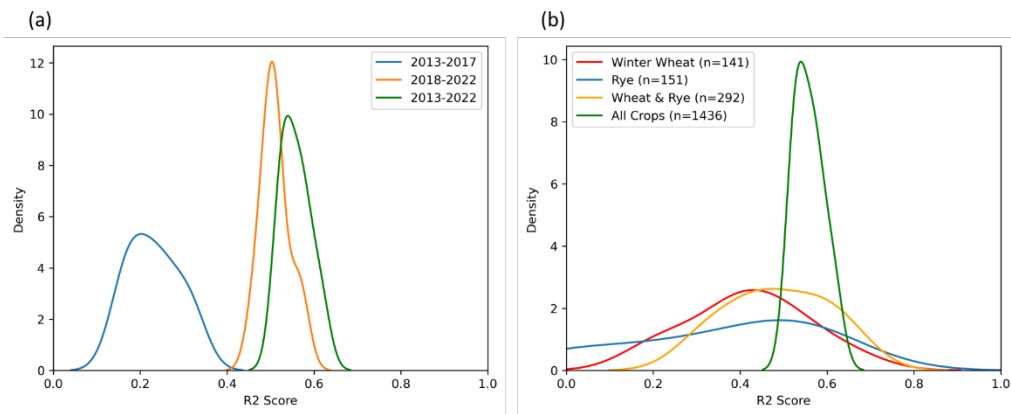

**Figure 13.** Distributions of the R² skill score based on 10 repetitions for each setup. (a) Separate models for pre-drought and drought years. (b) Separate models for individual crop types.


Model inspection identifies SPEI below -1 in June as the most relevant condition in the lumped model for all crops (Fig. 14) and also in crop-specific models for wheat and rye (Appendix D). Note that the features on county level always refer to the relative affected area above or below a threshold, e.g. the value of "SPEI June < -1" indicates the relative area per crop per
county affected by SPEI in June below -1. However, a large fraction of the data indicates that non-exceeding -1 coincided with negative empirical yield gaps, i.e. higher than expected yields. To investigate this in more detail, we run another model setup using only data with positive empirical yield gap (n=827). Data on county level always includes mixed effects, i.e. the constraint "empirical yield gap > 0" on county level does not imply that there are no damaged fields in the data, but rather that damaged fields are outweighed by fields with higher than expected yields within the same county. Features based on SPEI in
June are still among the most important predictors for such a subset, with thresholds of -0.5, and -1 ranked high (Fig. 14b). Even more severe meteorological drought conditions (SPEI < -2) are apparently just too rare in this dataset to be influential on county level. In March the threshold of 0 is again selected in reverse direction, i.e. indicating damage from too wet conditions (cf. Fig. 15d). Multiple AZL features are selected, confirming once more that soil quality is a relevant drought vulnerability factor (the regression target is already based on expected yield estimates that account for AZL, so this effect is on top).
LST/NDVI as predictive feature for the empirical yield gaps is of low relevance when using all data, but ranks higher when restricting the training data to positive yield gaps. In the comparison of crops (Fig. 15a), lupines clearly stick out, which is explained by the high losses in the yield data (cf. Fig. 7). The interaction of crop type with AZL < 36 shows once more that rye is growing on worse soils than wheat, but still has lower SHAP values with respect to the regression on impacts. Triticale is on a similar level as wheat, canola even higher. From all these crops, rye is thus empirically found the most robust.


To check the stability of the SHAP values, we repeated the model fitting several times and inspected the resulting summary plots. The first features are always crop type and SPEI in June. Beyond the first few ranks, feature effects get very similar and the exact ranks can shift in repeated model runs (depending on the random data subset and respective model parameters). The

effects for the different crop types and shapes of the dependence plots are also stable results, confirmed in multiple setups.
Focusing the models on positive empirical yield gaps can make the feature effects more linear (Fig. 15b and 15c). Nonlinear responses in the dependence plots for single features on county level are likely empirical artefacts, as the definition of a feature as relative affected area should more or less linearize the physical response. Although spatial neighbourhood effects, like water lacking in a hydrologically connected area, could introduce nonlinearities, we assume in general that more affected area should lead to more impact, regardless of the criterion.


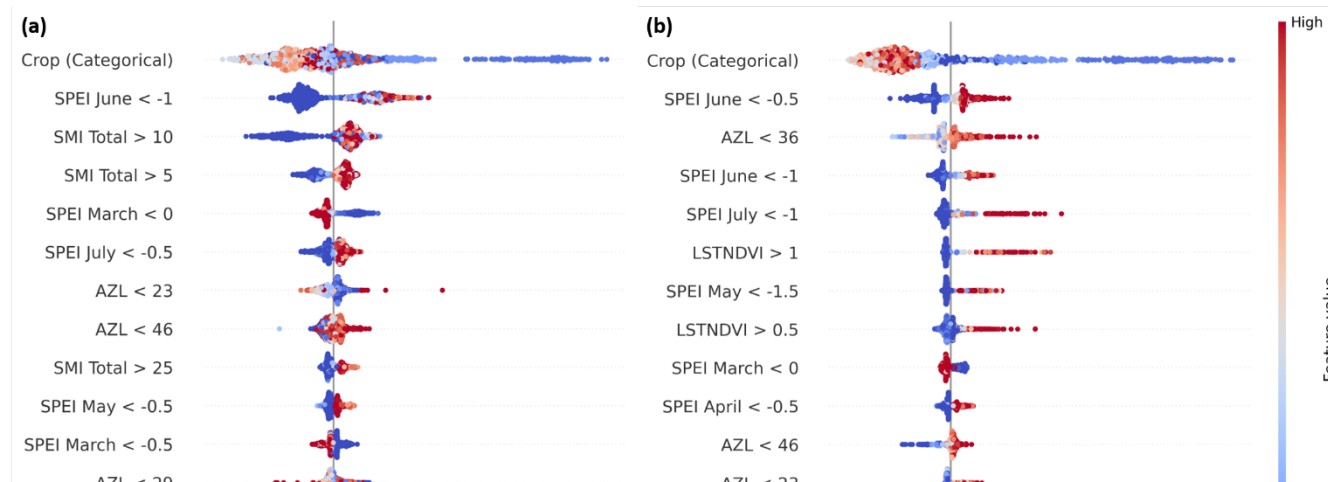

**Figure 14.** SHAP summary plots for the best model trained on (a) all data, and (b) empirical yield gap > 0

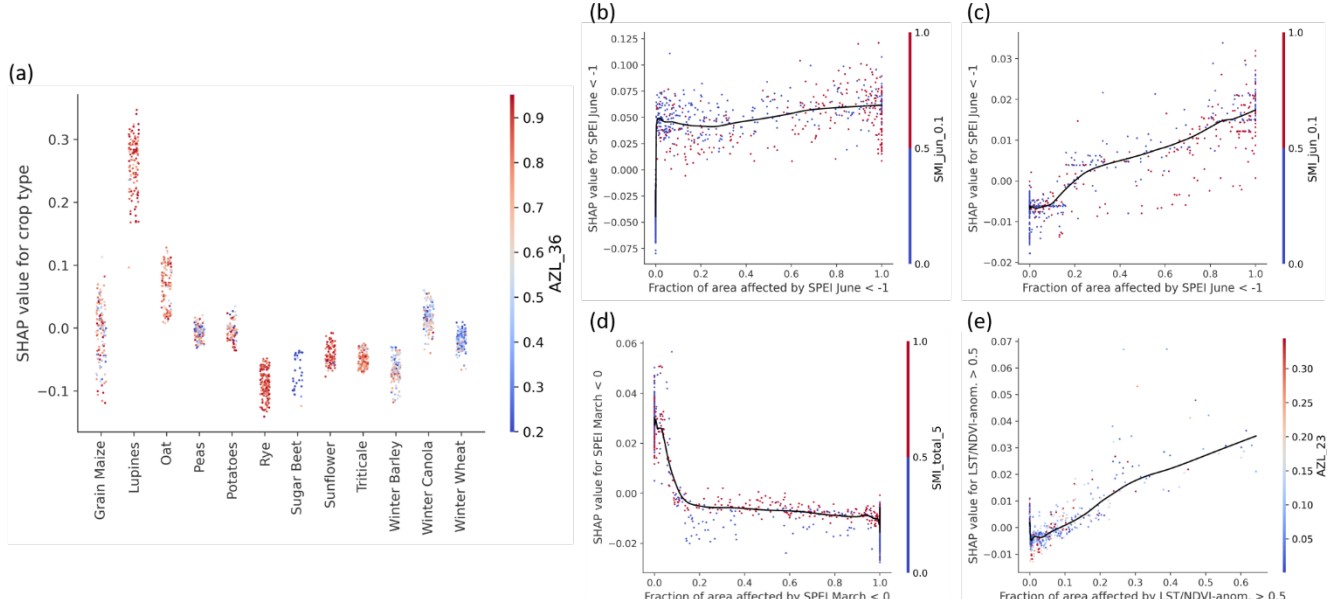

**Figure 15.** SHAP dependence plots for selected features on county level. (a) Effect of crop type and interaction with relative area of AZL < 36 from a model trained on all data (b) Effect of SPEI in June < -1 from a model trained on all data and (c) from a model trained only on positive empirical yield gaps. (d) Effect of SPEI in March < 0 from a model trained only on positive empirical yield gaps. (e) Effect of LST/NDVI anomaly > 0.5 from a model trained only on positive empirical yield gaps.

## 3.3 Summary discussion

### 3.3.1 Key findings

The main innovation of our study is the comparison of impact-relevant factors derived from field-level thermal-spectral ratios to those derived from county level yield gaps via consistent XAI methods. Anomalies of LST/NDVI are shifted to higher values during the drought years, but spatial patterns are rather scattered. The South-East of Brandenburg ranks high in our per-hectare economic loss estimates throughout all of the investigated years, although in the exceptional years 2018 and 2019 high losses are also registered in the North and West. It is not immediately obvious how the spatial patterns of the individual hazard and vulnerability indicators relate to both impact indicators. While other studies have already presented regression attempts for drought impacts on individual crops in Germany (Peichl et al., 2021), crops vs forest in Thailand (Tanguy et al., 2023), multiple sectors across Europe (Poljanšek et al., 2021; Rossi et al., 2023), or modelled economic loss under climate change scenarios (Naumann et al., 2021), none of these studies compared impact-relevant factors derived on field level and county level from different impact data sources via XGBoost and SHAP. Through this comparison, we find the importance of SPEI in June for regressing the observed impacts substantiated by multiple model setups: (1) On field level, regressing LST/NDVI-anom., the SHAP values of SPEI in June strongly increase below -1. (2) On county level, regressing empirical yield gaps, the relative area affected by SPEI < -1 is selected as most important predictive feature for a model trained on all data, as well as for crop-specific models (both wheat and rye). (3) Even when removing all data where empirical yield gap < 0, i.e. more yield

reported than expected, SPEI features from June still top the ranking, although several thresholds are selected (mainly -0.5 and -1). This is of particular concern as current regional climate simulations for Brandenburg project a shift in seasonal water balance and intensity of rainfall: more precipitation than today might arrive in winter, but rainfall during the summer months is expected to occur in shorter and more intense downpours, which implies a lower fraction of infiltration and longer times of amplified evaporative loss between the rain events (Jacob et al., 2014; Coppola et al., 2021; MLUK, 2023). While uncertainties of these projections are rather high for the case of summer precipitation, more robust projections of increasing summer dryness have been shown for surface soil moisture (Berg et al., 2017; Cook et al., 2018). We further identify too wet conditions in March as an impact-relevant factor, in agreement with Peichl et al. (2021).

SMI-Total adds complementary information to monthly SPEI. No real model improvement is obtained when using both SPEI and SMI monthly values, though. From the considered vulnerability factors, AZL (i.e. agricultural soil quality) is by far the most relevant one. There is a clear influence of AZL on LST/NDIV-anom., with vulnerability rising at AZL below about 35. LST/NDVI is, somewhat surprisingly, not a good predictor for the empirical yield gaps in our study. We thus advise caution when interpreting empirical results from a single impact indicator. AZL is also related to selected crop types. Most notably, wheat is grown on high quality soils, while rye predominantly on low to medium quality soils. While this already indicates that rye tolerates harsher conditions, we find empirically that rye on poor soil is still more robust under drought conditions in the region than wheat on good soil – based on both impact datasets. The cropped area of rye decreased by about 30% between 2013 and 2022 in Brandenburg, though, and the area for winter wheat increased by 19% in the same time. Such choices of crop types simultaneously affect exposure and vulnerability, and thus risk.

### 3.3.2 Limitations & future research

From the monthly hazard features, the models can learn interactions that resemble accumulation – however, we did not include predictors from a previous year or even longer lag times. The only information on longer time is the SMI-Total (Fig. 8 shows the lag of 1 year compared to SPEI). As agricultural crops, as opposed to e.g. trees, are replaced every season, it does not seem logical to include longer lag times, but future research might investigate this. Groundwater and streamflow indicators have not been used, as both are highly managed in Brandenburg, and at the same time irrigation is very limited (as confirmed by personal communication with local experts), but we acknowledge that Rossi et al. (2023) found streamflow indicators relevant in the case of agriculture across Europe. Further improvements in modelling observed impacts likely require more detailed spatially explicit data on vulnerability, land use change, landscape organization, e.g. hedgerows, agroforestry systems, and (farm)land management, e.g. cover crops, fertilizer use, and irrigation. Agriculture in Brandenburg is predominantly rainfed, and we found no reliable spatially explicit dataset on irrigation. This gap could in the future be close via remote sensing studies. Most socio-economic variables used in our study, and in general in drought-related vulnerability studies (e.g. Meza et al., 2019; Stephan et al., 2023), might not exhibit direct influence on crop loss, but rather on the propagation of indirect impacts further down the

impact chain. Substantiating such theoretical assumptions with quantitative investigations is an important topic for future research, that requires novel datasets and methods, e.g. from the field of socio-hydrology (Wens et al., 2019)


The choice of impact variables, and preprocessing thereof, might introduce biases. LST/NDVI anomaly is a commonly used indicator for drought-related crop health, but others are possible, such as the radar vegetation index (Kim et al., 2012), hyperspectral metrics (Dao et al., 2021), fractional cover time series (Kowalski et al., 2023), or multimodal techniques (Karmakar et al., 2024). Regressions on county level are based on relative yield gaps. Although we did not identify rapid

agrotechnological changes within the investigated 10 years of yield data, the methodology could be improved to account for such potential jumps, particularly when investigating a longer time series. Directly regressing economic loss would also be possible, and lead to different insights (e.g. on the effect of price shocks). Both impact variables used in our regression are continuous rather than binary, which could affect the nonlinearities captured by the models.

We chose the algorithm XGBoost, which, compared to Random Forest, limits the amount of variability between the individual decision trees. This is assumed to avoid erratic behavior, but on the other hand could also limit the potential damaging processes discovered by the models. For the models on county level, predictive features were derived by computing the relative area above/below evenly-spaced thresholds. An alternative here would be to use quantiles, or to automate the feature engineering by deep learning algorithms. Stronger AI methods, not only in the regression but also in the feature learning step (i.e. deep

learning), could improve the predictive skill. While the $R^2$ scores obtained by our models are in range of similar studies (e.g. Peichl et al., 2021; Tanguy et al., 2023), they are still rather low for a predictive use case (which was not our aim in this study). Reasons for this often low to moderate model skill of such studies include uncertainty in the regression target, spatial and temporal resolution of the predictors, missing predictors and/or imperfect feature engineering, lack of representative training samples covering the entire nonlinearities and interactions in the natural processes, among others.


### 3.3.3 Recommendations

To prepare the agricultural sector, rural population and society for the uncertain future climate with an increased frequency of extreme hydrometeorological events, monitoring systems with early warnings are needed. Given that most decision makers, e.g. local authorities, disaster managers, or farmers, react to information about impacts (Dutt & Gonzales, 2010), such

monitoring and early warning systems should be impact-based, rather than only inform about hazard. In particular we recommend to

1. Foster the implementation of impact-based monitoring and early warning systems for droughts to reduce impacts
2. Establish the use of interactive visualization tools in education and training to advance adaptation

3. Select drought-robust crops (farmers), e.g. rye over wheat; avoid adverse incentives (policy makers)
4. Provide water storage or other capacities for ad-hoc measures during the decisive summer months (here: June)

## 4. Conclusion

Our analysis of spatiotemporal patters of agricultural drought hazard, exposure, vulnerability, and impact indicators for Brandenburg, 2013-2022, empirically shows that the links between these components are complex and, consequently, risk mapping and monitoring need to be supported by thorough investigations from multiple datasets. We present agricultural impact indicators on two spatial levels – the crop health indicator LST/NDVI on individual fields, and empirical yield gaps on county level – and apply XGBoost regression to relate both of them to hazard and vulnerability indicators. Finding more detailed data on vulnerability and farmland management is still challenging, but supposedly needed to improve the skill of the models. Stronger remote sensing indicators on drought impacts, beyond LST/NDVI, seem necessary as well. Data-driven techniques from the AI domain can capture complex interactions in human-environments such as agriculture. SHAP plots uncover which factors drive the prediction of impact indicators in the models. This does not necessarily relate to causal effects in nature, though. We thus suggest to cross-check results obtained from different model setups, different regression targets, and ideally also different algorithms. Model inspection in this study shows that features are generally used in a physically meaningful direction, which is a prerequisite if data-driven models are to be trusted. Models from both impact datasets agree on the importance of meteorological drought in June, soil quality, and the type of crop. No single feature explains the full data, though, and in fact such simplified interpretations are against the logic of using a strongly nonlinear ML algorithm to tackle complex regression problems. Rather than attempting to weight indicators manually, empirical impact data should be the benchmark to evaluate hazard and vulnerability indicators for the purpose of risk mapping. Interactive visualization tools should enter the education system at all levels to train risk and climate literacy of future citizens, and demonstrate impacts of hazards rather than hazards only. Ultimately, interactive impact-based forecasting tools would offer a basis for science communication with policy makers and participatory modelling approaches to develop better climate policies and raise awareness for feasible adaptation options.

**Acknowledgements**:

This research was supported by the Einstein Research Unit "Climate and Water under Change" from the Einstein Foundation Berlin and Berlin University Alliance (ERU-2020-609), the Deutsche Forschungsgemeinschaft (DFG, German Research Foundation) – Research Unit 2569, 'Agricultural Land Markets—Efficiency and Regulation' and SFB 1502/1–2022 - project number: 450058266. We thank Jan Sodoge for providing the newspaper text-mining data for the full investigated period of time, and acknowledge the preliminary work conducted by Thomas Hoffmann and Marlen Laudien during their theses at HU Berlin. We are also very thankful to all data providers. We further thank both reviewers for adding important points to the discussion.

## Author contribution

FB(1): Conceptualization, Methodology, Software, Formal analysis, Validation, Visualization, Writing–original draft. PA: Resources, Writing–review & editing. HZ: Resources, Writing–review & editing. FB(2): Resources, Writing–review & editing. SH: Resources, Writing–review & editing. TL: Conceptualization, Resources, Writing–review & editing, Funding acquisition. All authors have read and agreed to the submitted version of the manuscript.

## Competing interests

PA is member of the editorial board of this special issue. All other authors declare that they have no conflict of interest.

## Code and data availability

All data and scripts needed to reproduce the figures, as well as the full processed dataset and scripts used to conduct the preprocessing and analysis are publicly available via GitHub: https://github.com/fabiobrill/brandenburg-drought-study/ and permanently archived on Zenodo: https://doi.org/10.5281/zenodo.13373271. Except for the crop prices, which were obtained from AMI under a commercial license, all other raw data used in this study are either open (see Table 1) or can be made available upon reasonable request to the authors. The interactive data exploration app in R-Shiny is also available via the GitHub repository and can be run locally. An independent publicly hosted version is accessible online: https://fabiobrill.shinyapps.io/agrdrought-explorer-brandenburg/

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

## Appendix A

**Table A1.** Average yields [dt/ha] per LBG 2010–2014, as used to estimate expected yields. Compiled from LELF (2016)

| Crop | LBG-1 | LBG-2 | LBG-3 | LBG-4 | LBG-5 |
|---|---|---|---|---|---|
| Winter wheat | 77 | 65 | 50 | 38 | 23 |
| Winter rye | 63 | 55 | 43 | 35 | 25 |
| Summer rye | 37* | 33* | 25.8* | 21* | 15* |
| Winter barley | 75 | 63 | 50 | 36 | 25 |
| Oat | 55 | 45 | 35 | 27 | 18 |
| Winter triticale | 66 | 60 | 48 | 37 | 23 |
| Summer triticale | 39.6* | 36* | 28.8* | 22.2* | 13.8* |
| Grain maize | 90 | 80 | 70 | 60 | 50 |
| Peas | 35 | 30 | 25 | 20 | NA |
| Lupines | NA | 25 | 21 | 18 | 15 |
| Potatoes | 370 | 350 | 320 | 250 | 220 |
| Potatoes (starch) | 450 | 420 | 390 | 320 | 250 |
| Sugar beet | 650 | 620 | 580 | NA | NA |
| Winter canola | 43 | 38 | 32 | 25 | 20 |
| Summer canola | 23 | 18 | 14 | 11 | NA |
| Sunflower | 28 | 25 | 20 | 17 | 15 |

*Assumption, based on 60% of winter variety.

**Table A2.** Merging of the crop types between the three datasets IACS, yield reports, and average yields per LBG. Silage maize has been

discarded later, and also for sugar beet we did not find prices 2021-2022

| Crop | LBG average yields | IACS data | Yield reports | Assumptions made |
|---|---|---|---|---|
| Grain maize | Grain maize | Grain maize | Grain maize | - |
| Sunflower | Sunflower | Sunflower | Sunflower | - |
| Sugar beet | Sugar beet | Sugar beet | Sugar beet | - |
| Lupines | Lupines | Lupines | Lupines | - |
| Peas | Peas | Peas | Peas | - |
| Winter barley | Winter barley | Winter barley | Winter barley | - |
| Winter canola | Winter canola | Winter canola | Winter canola | - |
| Oat | Oat | Winter oat, Summer oat | Oat | Merge IACS to "Oat" |
| Potatoes | Potatoes Potatoes (starch) | Potatoes (various) Potatoes (starch) | Potatoes combined | Merge to "Potatoes" |
| Winter wheat | Winter wheat | Winter wheat | Winter wheat + spelt | Neglect spelt |
| Rye | Winter rye | Winter rye Summer rye | Rye + winter mix | Assume LBG values for summer rye as 60% of winter rye; Merge IACS to "Rye"; Neglect winter mix |
| Triticale | Winter triticale | Winter triticale Summer triticale | Triticale | Assume LBG values for summer triticale as 60% of winter triticale; Merge IACS to "Triticale" |

## Appendix B

**Table B**. Intervals for thresholds

| Indicator category | Interval for thresholds (exact values) |
|---|---|
| SPEI | 0.5 (-4*., -3.5*, -3*, -2.5, -2, -1.5, -1, -0.5, 0) |
| SMI | 0.05 (0, 0.05, 0.10, 0.15) |
| SMI-Total | 5 (0, 5, 10, 15, 20, 25, 30, 35) |
| LST/NDVI-anom. | 0.25 (0, 0.25, 0.50, 0.75, 1.00, 1.25, 1.50) |
| AZL | LBGs (23, 29, 36, 46) |

*only for SPEI-Magnitude

## Appendix C

**Table C1.** Model setups on field level (y = LST/NDVI-anom.). The indicators denoted with an 'x' are included in the respective setup. Performance initially assessed on 10% of the data to check the relative differences.

| Setup | Crop type | SPEI Magnitude | SPEI Monthly | SMI Magnitude | SMI Monthly | Total Soil Magnitude | Vulnerability AZL, TWI, nFK | $R^2$ (mean of 10 repetitions) |
|---|---|---|---|---|---|---|---|---|
| F1 | | x | | x | | x | | 0.09 |
| F2 | x | x | | x | | x | x | 0.17 |
| F3 | x | | x | | | | | 0.20 |
| F4 | x | | | | x | | | 0.15 |
| F5 | x | | x | x | | | x | 0.26 |
| F6 | x* | | x | x | | | x | 0.25 / 0.48** |
| F7 | x* | | x | | x | x | x | 0.25 / 0.51** |

*as categorical feature rather than one-hot encoded, ** re-trained on the full dataset

**Table C2.** Model setups on county level (target = relative empirical yield gap) using all available samples per setup (scores on holdout data). The indicators denoted with an 'x' are included in the respective setup.

| Setup | Crop type | LST/NDVI | SPEI Magnitude | SPEI Monthly | SMI Magnitude | SMI Monthly | SMI Total | Vulnerability AZL | $R^2$ (mean of 10 repetitions) |
|---|---|---|---|---|---|---|---|---|---|
| LK1 | x | x | | | | | | | 0.22 |
| LK2 | x | | x | | | | | | 0.41 |
| LK3 | x | | x | | x | | x | x | 0.52 |
| LK4 | x | | | x | | | | x | 0.54 |
| LK5 | x | | | | | x | | x | 0.48 |
| LK6 | x | | | x | | x | | x | 0.53 |
| LK7 | x | | | x | | x | x | x | 0.56 |
| LK8 | x | x | | x | | x | x | x | 0.57 |
| LK9 | x* | x | | x | | x | x | x | 0.53 |
| LK9b | x* | x | | x | | x | x | x | 0.40** |

*as categorical feature rather than one-hot encoded, **trained only on samples where empirical yield gap > 0


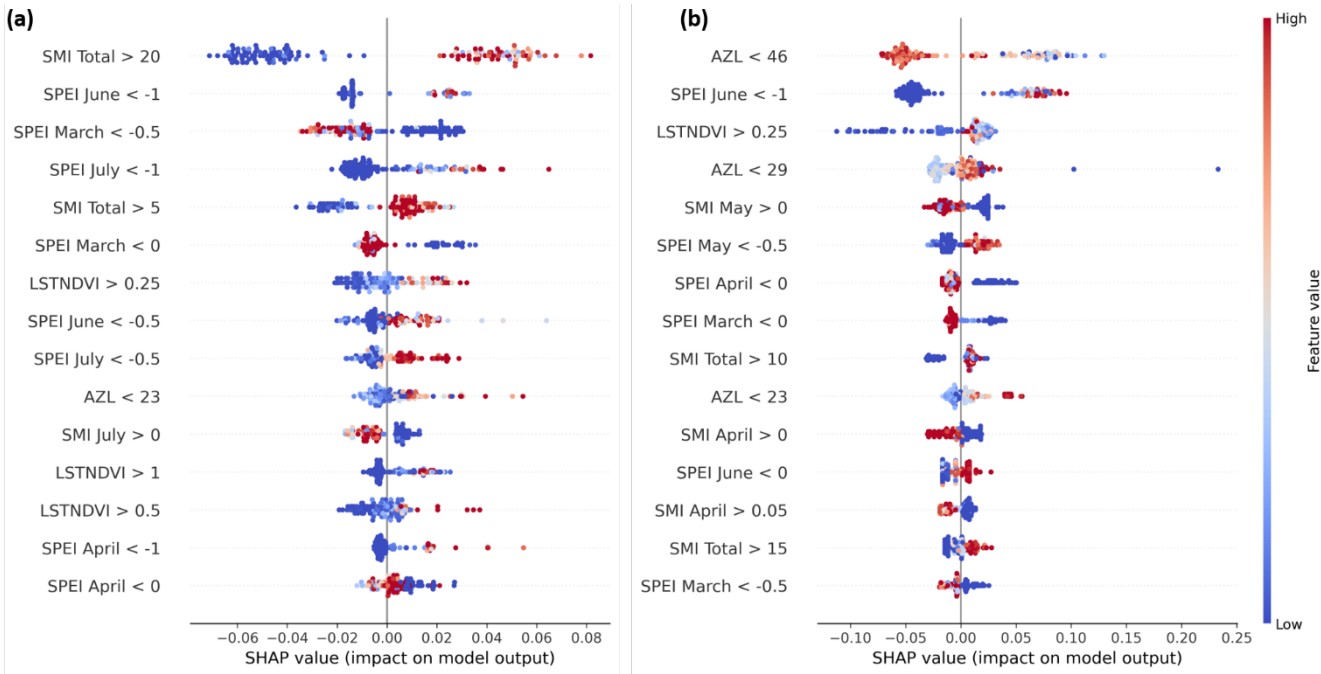

**Figure D1**. SHAP summary plots for models trained only on (a) wheat (b) rye

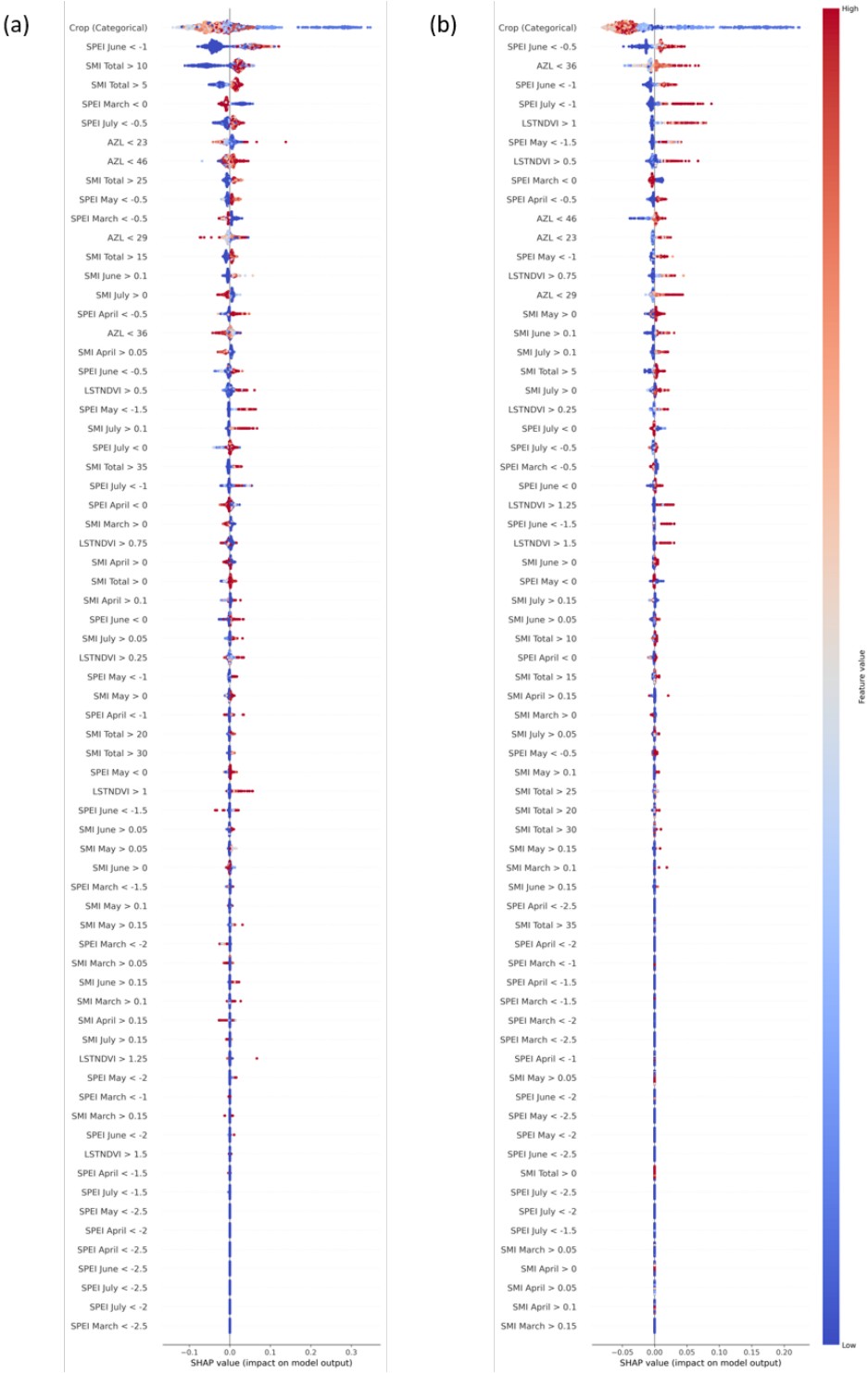


**Figure D2**. SHAP values for all features of the best model trained on (a) all data, and (b) empirical yield gap > 0.
Fig. 14 in the main paper only displays the first 15 of these.