# Peer review of "Exploring drought hazard, vulnerability, and related impacts to agriculture in Brandenburg"

_EGUsphere, 2024_

## Referee Comment (RC2)

**Reviewer's comments: manuscript egusphere-2024-1149**

In this study, the authors used a multi-index approach (exposure, risk, vulnerability) to model the impacts of drought events on agricultural systems in the German federal state of Brandenburg, considering the LST/NDVI ratio as the response variable. The scientific approach used is valid. It reflects the multifactorial complexity of the implications of drought for the productivity of the region's farming systems. However, there are a few points of clarification, particularly in the methodology section. My comments and remarks are as follows:

**Comment 1**

« Empirical associations to the impact indicators on both spatial levels are compared. Non-linear models explain up to about 60% variance in the yield gap data, with lumped models for all crops being more stable than models for individual crops». This is imprecise, you must specify the names of the nonlinear models as well as for the grouped models. It is also important to include in the abstract the performance statistics of the models used.

**Comment 2**

"Rye is found less vulnerable than wheat, despite growing on poorer soils". The fact that rye grows on poorer soils is a proof that it is more resilient and less vulnerable than wheat, so I do not see why the conjunction of subordination although? The sentence was rephrased.

**Comment 3**

In introduction, « This has implications for modelling and Monitoring ». You mean implications in the modelling and monitoring of agricultural drought. If so, the sentence should be completed.

**Comment 4**

Overall, the introduction is well written and argued. However, the application of artificial intelligence models in modelling drought impacts, risk, and vulnerability has been limited. It is worth adding a paragraph on the advantages and limitations of intelligence models in modelling the impacts of drought given that in your methodology you have used the extreme gradient boosting algorithm (XGBoost).

**Comment 5**

*Line 250* «To retain as much information about the hazard distributions, we computed the relative affected area (non-)exceeding specified thresholds (in regular intervals of 0.5 for SPEI, 0.25 for LST/NDVI-anom., 0.05 for SMI, 5 for SMI-Total, and using the LBG class limits for AZL). A total of 68 features were created this way on county level».

On what criterion were these thresholds considered? This deserves to be clarified. The different classification thresholds for these indices and their meanings should be provided in a table in the methodology section.

**Comment 6**

The principle of the calculation of the LST/NDVI anomaly has not been sufficiently described. There should be a separate section to better describe and justify the choice of this anomaly to represent the impacts of drought when there are various other anomalies or indices that can better reflect the impacts of drought on agricultural systems. In this sense, the normalization indicated in Table 1 concerns only the LST values and/or the LST/NDVI values. If so, considering the max and min values or mean and standard deviation (SD)?

**Comment 7**

Ligne 255-260 « In 2013 and 2014 the SMI-Total is close to 0, observed vegetation health is at its maximum (i.e. negative LST/NDVI-anom.), essentially no impact-related statements…..» Similarly, to better assess the consistency of these statements, the formula and principle of the calculation of the IMS and IMS-Total must be clearly described in the methodological section with the different classification thresholds.

**Comment 8**

In Table 1, you mentioned that the monthly SPEI used has a resolution of 10 km and the source is the reference Zhang et al. (2024). However, in this reference, the SPEI used has a 1 km resolution. It is a bit ambiguous. Has the SPEI been calculated? or was the same database from the Zhang et al. (2024) study used? If this is the case, the spatial resolution of 10 km should be rectified because in the source reference mentioned it is rather 1 km that is mentioned.

**Comment 9**

The algorithm used to calculate the Landsat LST was not explained in the methodology.

---

## Author Response (AR1)

We would like to thank both Reviewers and the Editor for the overall positive evaluation of our work, and the constructive feedback. In the following, we respond to all raised comments individually. The responses are numbered, with the first digit indicating the reviewer, and the following digits indicating the order of the comments. We are confident that our thoroughly revised manuscript now addresses all raised points and hope for it to be considered worthy of publication in NHESS.

Line numbers refer to the "tracked changes" version of the manuscript.

| 1. | **Reviewer comment** |
|---|---|
| | The manuscript and research behind are highly interesting, innovative and relevant to the journal. I have only very few comments, except that I miss a discussion section (I don't find a deep gap analysis, recommendations for future research nor a comprehensive summary of findings with a thorough comparison with existing research outputs in the current version - parts of it are covered in other chapters but I don't think that is clear enough). Besides, While the results are written down neatly with some informative figures, it is hard to follow for people not working with similar models. I think the manuscript could benefit from a sentence here and there saying "meaning that..." where the result is explained in an easily interpretable way (especially in the parts where SHAP is used). |
| | **Author's response** |
| | We would like to thank the reviewer for the overall positive evaluation of our work. Regarding the discussion section, we originally decided for a combined "Results & Discussion" section because of the complexity of the results, which implies that the discussion parts are scattered across the manuscript. This also applies to the gap analysis and comparison with existing research. For this reason, we used the "Conclusion" chapter for a slightly longer summary. However, in response to this reviewer comment, we now include a separate subchapter "Summary discussion" at the end of the "Results & Discussion" chapter, with the subheadings "Key findings", "Limitations & future research", and "Recommendations". At the same time, we considerably shortened the "Conclusion" and moved parts from this section up to the new "Summary discussion".

Following your suggestion, we also checked the manuscript and added additional explanations to make the text easier to read and follow. |
| | **Changes in the manuscript** |

[revised manuscript text omitted]

L399: ", meaning that vulnerability is higher on soils below that quality"

| 1.01 | **Reviewer comment** |
|---|---|
| | There is a clear justification of the research and methodological choices made. While referenced once, the method is quite like the study of Naumann et al 2021 and the European Drought atlas - the differences can be highlighted better. |
| | **Author's response** |
| | Thank you for your comment. We agree that there are some similarities to the methodology used in the European drought atlas, which we already cited, and other works where G. Naumann was involved (e.g. Poljanšek et al. 2021). Naumann et al. 2021 is a very interesting paper with a slightly different scope, though. We added this reference to the introduction. |
| | Our study differs substantially in the methods and the level of detail. Compared to European level studies, we dive deeper into regional details of Brandenburg, which is addressing the scope of the special issue. In particular, our comparison of field-level resolution thermal-spectral measurements and reported yield statistics via XGBoost & SHAP is a unique point. To the best of our knowledge, no other published study presents such a comparison. We have now added a paragraph to highlight differences of our approach to the mentioned literature in the discussion chapter. |
| | We introduced a new subchapter, "3.3.1 key findings", in which we point this out |
| | **Changes in the manuscript** |
| | L66: „and severe increases of economic impacts are projected for climate change scenarios without adaptation (Neumann et al. 2021) " |
| | L501: "3.3.1 Key findings
The main innovation of our study is the comparison of impact-relevant factors derived from field-level thermal-spectral ratios to those derived from county level yield gaps via consistent XAI methods. (…) While other studies have already presented regression attempts for drought impacts on individual crops in Germany (Peichl et al., 2021), crops vs forest in Thailand (Tanguy et al., 2023), multiple sectors across Europe (Poljanšek et al., 2021; Rossi et al., 2023), or modelled economic loss under climate change scenarios (Naumann et al., 2021), none of these studies compared impact-relevant factors derived on field level and county level from different impact data sources via XGBoost and SHAP. Through this comparison, we find (…)" |

| 1.02 | **Reviewer comment** |
|---|---|
| | I like that multiples ways of looking at (quantifying) impact are tested, that you compare empirical and modelled impact on production. The general workflow figure is very clear. |
| | **Author's response** |
| | Thank you |
| | **Changes in the manuscript** |
| | |

| 1.03 | **Reviewer comment** |
|---|---|
| | In line 141, I would disagree with the definition of vulnerability (or the phrasing thereof) as a characteristic of exposure. Maybe as an internal characteristic of the exposed items? At least the IPCC would not describe it that way. |
| | **Author's response** |
| | Thank you for this observation. We admit that this embedded sentence was rather confusing and not 100% precise, although it was intended to be in line with the IPCC definitions (our first author, FB, strongly supports the language guidance by Reisinger et al., 2020, which is cited in the third sentence of our manuscript, line 38). We removed the unnecessary and confusing part, and rephrased the following sentence. |
| | **Changes in the manuscript** |
| | L149: "Vulnerability indicators attempt to capture the relevant characteristics that shape the relationship between hazard intensity and impacts." |

| 1.04 | **Reviewer comment** |
|---|---|
| | I wonder why a groundwater and/or streamflow indicator was not considered as potential hazard/predictor? And I like the calculating of the magnitude of deficit, I wonder how sensitive the results are to the choice of -0.5 as threshold for these? |
| | **Author's response** |
| | Groundwater and streamflow in Brandenburg are highly managed, and at the same time use for irrigation is very limited. Obtaining reliable interpolated data on groundwater is also not trivial, and a current topic of investigation (Somogyvári et al., 2024). We hope to cover the connection to deep soil water via the total soil drought magnitude. This being said, we agree that further indicators could have been included. It is true that Rossi et al. did use streamflow indicators also for agricultural impacts, and found it somewhat important in the models – however, we interpret this as primarily related to irrigated agriculture and/or Southern European contexts. We added this point to the new subchapter "Limitations & future research"

A threshold of -0.5 on SPEI to characterize drought is used for example by Wang et al. (2014, 2021). Sometimes -1 is found in other literature. Please note that this choice only slightly affects one particular indicator, which is used in the descriptive figures 4 and 8, but does not play a role in the machine learning part. The aggregated magnitude does not provide additional information over the monthly data, as XGBoost can internally compile any sort of accumulated features from the monthly |

layers. It has not been used in the evaluated model runs as documented in Appendix B. We added the reference to Wang et al. (2021)

*Somogyvári, M., Brill, F., Tsypin, M., and Krueger, T.: A top-down modeling approach to assess regional scale groundwater vulnerability: a case study for Berlin-Brandenburg, EGU General Assembly 2024, Vienna, Austria, 14–19 Apr 2024, EGU24-16699, https://doi.org/10.5194/egusphere-egu24-16699, 2024.

| **Changes in the manuscript** |
|---|
| L542: "Groundwater and streamflow indicators have not been used, as both are highly managed in Brandenburg, and at the same time irrigation is still quite limited (as confirmed by personal communication with local experts), but we acknowledge that Rossi et al. (2023) found streamflow indicators relevant in the case of agriculture across Europe."

L178: "(cf. Wang et al. 2021 for SPEI thresholds)." |

| 1.05 | **Reviewer comment** |
|---|---|
| | The detrending of the impact data is done with a moving window: were there no sharp agrotech jumps in the yield over time? |
| | **Author's response** |
| | We did not observe such clear jumps in the yield data during the investigated 10 years – which is still a rather short time for agrotechnical development in an already industrialized country. However, identifying such effects was also not our primary interest, so there is a possibility of undetected effects that might negatively affect the regression skill. In response to this comment we added to the discussion chapter that one limitation is that we did not account for sudden changes in the yield over time. |
| | **Changes in the manuscript** |
| | L557: "Although we did not identify rapid agrotechnological changes within the investigated 10 years of yield data, the methodology could be improved to account for such potential jumps, particularly when investigating a longer time series." |

| 1.06 | **Reviewer comment** |
|---|---|
| | L193: "we refer to..." this sentence is a bit unclear. |
| | **Author's response** |
| | We rephrased the sentence |
| | **Changes in the manuscript** |
| | L216 The line now reads: "The empirical yield gap divided by the expected yield is called "relative gap"" |

| | |
|---|---|
| **1.07** | **Reviewer comment** |
| | Paragraph starting at L203: it is a bit unclear whether you take modelled or empirical yield gaps as closer to 'the reality'. Also, starting from line 209, this alinea is fuzzy. i think this is the first time there is a reference to a reference period? I don't fully understand what is conveyed there - maybe rephrase? |
| | **Author's response** |
| | Our "expected yield" estimates are based on the pre-drought years, as described in the respective subchapter (now 2.4.2 in the revised manuscript). If this pre-drought period is a good reference, values of the expected yield should thus be close to potential production. The process model WOFOST is used for a cross-check, although it is a global model and we assume the empirical approach to be closer to the actual local conditions. In particular, WOFOST does not account for the different soil quality ranges (LBG), and the figure clearly shows that there is a spread in the empirical data, depending on the soil quality range, while WOFOST always assumes a rather good soil (LBG-2 or LBG-1). We rephrased and added more details as well as a new subheading |
| | **Changes in the manuscript** |
| | L226: The rephrased paragraph with new subheading now reads

"2.4.3 Comparison to external data
For a plausibility check, we compared the resulting empirical yield gaps and loss estimates to regional newspaper reports. For individual crops (rye, wheat, maize, barley) we were able to additionally calculate the potential production (PP) and water-limited production (WLP) by the process model WOFOST on a 2 km grid resolution (Jänicke et al., 2017; de Wit et al., 2019). If our expected yields from the pre-drought years are realistic, they should be similar to the potential production. Crop growth in WOFOST is modelled from irradiation, temperature, $CO_2$ concentration, plant characteristics, seeding date, and availability of water. The physically modelled potential production from this simulation matches very well with the expected yields derived by our empirical approach for soil quality range LBG-2 in the case of wheat and barley, and LGB-1 in the case of rye (Fig. 3). We are thus confident that our approach produces estimates in a realistic range. Only for maize the modelled potential production is higher than the average values for Brandenburg suggest on any soil type. This comparison also underlines that it is important to account for the soil quality range, and thus our empirical approach appears more realistic than this particular WOFOST simulation. For further comparison we use the newspaper reported impact score by (Sodoge et al., 2023), for the category "agriculture". All data used is summarised in Table 1." |

| | |
|---|---|
| **1.08** | **Reviewer comment** |
| | Looking into the list of indicators, I would miss some related to irrigation and general farm management, county rules on when crops can be planted/harvested, use of fertiliser, market prices etc. Some of this info might be available? |
| | **Author's response** |
| | Thank you for your critical evaluations and suggestions for additional indicators. Indeed, we thought about using these, too. Although agriculture in Brandenburg is predominantly rainfed, we intended to include irrigated area - but figured out that there is no reliable spatially explicit dataset for the region. A study on remote sensing based irrigation mapping is currently in preparation by a colleague, but their results were not available at the time when this study was conducted. As for farm management |

and fertilizer use, we are not aware of (openly available) data on the level of counties or below either. We added sentences on future research recommendations.

Irrigation and more detailed data on vulnerability and management were already requested in the manuscript conclusion in L590ff: "Further improvements in modelling observed impacts likely require more detailed spatially explicit data on vulnerability and management, e.g. irrigation"

Market prices have been included in the economic estimate, but not for the regression. The regressions on county level are based on the relative yield gaps, i.e. fractions of crops assumed lost, as this is arguably the last "physical" observable impact. Directly regressing the economic loss would introduce further uncertainties. It is a common procedure in risk modelling to regress the physical impact and then combine this with price data afterwards (e.g. Merz et al., 2010; Sairam et al., 2020)

| **Changes in the manuscript** |
| --- |
| L545: "Further improvements in modelling observed impacts likely require more detailed spatially explicit data on vulnerability, land use change, landscape organization, e.g. hedgerows, agroforestry systems, and (farm)land management, e.g. cover crops, fertilizer use, and irrigation. Agriculture in Brandenburg is predominantly rainfed, and we found no reliable spatially explicit dataset on irrigation. This gap could in the future be close via remote sensing studies." |
| |
| L614: Finding more detailed data on vulnerability and farmland management is still challenging, but supposedly needed to improve the skill of the models |

| 1.09 | **Reviewer comment** |
| --- | --- |
| | (most of) the socio economic vulnerability indicators will barely have an effect on the hazard impact link (if impacts are yield deficits) but will influence how this drought loss cascades through society. A critical reflection could be good here. |
| | **Author's response** |
| | Yes, this is true, and we debated this issue among the co-authors during the design of the study. As a result of our internal debate, and due to availability and resolution of data, the study focus is stronger in the biophysical aspects of risk. We made this more clear in the discussion section. |
| | **Changes in the manuscript** |
| | The paragraph now reads: |
| | "Most socio-economic variables used in our study, and in general in drought-related vulnerability studies (e.g. Meza et al., 2019; Stephan et al., 2023), might not exhibit direct influence on crop loss, but rather on the propagation of indirect impacts further down the impact chain. Substantiating such theoretical assumptions with quantitative investigations is an important topic for future research, that requires novel datasets, though." |

| 1.10 | **Reviewer comment** |
| --- | --- |
| | l324: this paragraph is raises some questions. how does it relate to the previous paragraphs? Why is this relevant / what is the key take away from it? |
| | **Author's response** |
| | Thank you for this observation. We removed the detached paragraph |

| | **Changes in the manuscript** |
|---|---|
| | L356: Removed "~~A number of socio-economic vulnerability indicators are particularly concerning in the North-Western areas Prignitz and Ostprignitz-Ruppin, as well as in the Southern county Oberspreewald-Lausitz: those regions rank above average on agricultural dependency for livelihood and below average on secured succession, while Prignitz has a particularly high agricultural population density on top. All three exhibit low scores for the coping capacity indicators education and participation in local politics~~" |

| 1.11 | **Reviewer comment** |
|---|---|
| | The R2 scores are not high. It is explained in the manuscript, but some figures showing time series of obs/pred could help explaining why that is not considered problematically low. and add in the discussion how this could potentially be improved. |
| | **Author's response** |
| | While we agree with the reviewer that the R2 scores are not very high, they are still in line with similar published attempts, as cited in our manuscript (e.g. Peichl et al. 2021, Tanguy et al. 2023). Reasons for this often low to moderate model skill of such studies include uncertainty in the regression target, spatial and temporal resolution of the predictors, missing predictors and/or imperfect feature engineering, lack of representative training samples covering the entire nonlinearities and interactions in the natural processes, among others. We added further suggestions on how to potentially improve the model fit.

We apologize for not fully understanding the request about figures of time series of obs/pred. A time series would imply that data is subdivided into the 10 individual years, i.e. 10% in each split. While the data on field level would be sufficient for such splits, the data on county level appears too small for this to be meaningful. As we trained many models, producing such a plot is not straightforward, and might not add much value to the overall narrative of the manuscript. Figure 13 in our manuscripts shows the distribution of skill for repeated training of different setups on county level, and the effect of merging or subdividing the training data. In particular, Figure 13a shows the considerably lower skill for the years 2013-2017, and the improvement when sampling training data across all years (note once more that 10 years is still a short time span for a machine learning task). This is more information on the model skill than typically reported by similar studies (e.g. Rossi et al. do not report skill at all, and still interpret the occurrence of features in decision tree models, although you might have more information on this than we have)

Further, we would like to point out that predictive modelling was not the main purpose of our study, although we advocate for the development of such models. If producing an operational prediction model were the aim, we might choose a different temporal training/test strategy and present additional skill scores that provide insights into the particular types of predictive errors. In that case we might also employ a modelling framework that quantifies uncertainty. Our primary aim here was to identify impact-relevant factors from the data, and compare the results obtained from two different impact datasets via the same methods. The knowledge gained might in the future be used to construct sparse predictive models, but we consider this out of scope for this particular publication. |

| | **Changes in the manuscript** |
|---|---|
| | L567: "Stronger AI methods, not only in the regression but also in the feature learning step (i.e. deep learning), could improve the predictive skill. While the R² scores obtained by our models are in range of similar studies (e.g. Peichl et al., 2021; Tanguy et al., 2023), they are still rather low for a predictive use case (which was not our aim in this study). Reasons for this often low to moderate model skill of such studies include uncertainty in the regression target, spatial and temporal resolution of the predictors, missing predictors and/or imperfect feature engineering, lack of representative training samples covering the entire nonlinearities and interactions in the natural processes, among others."

 L614: "Finding more detailed data on vulnerability and farmland management is still challenging, but supposedly needed to improve the skill of the models. Stronger remote sensing indicators on drought impacts, beyond LST/NDVI, seem necessary as well." |

| 1.12 | **Reviewer comment** |
|---|---|
| | The paragraph starting at l403 is a nice and critical piece. also the conclusion is concise, comprehensive and clear |
| | **Author's response** |
| | Thank you |
| | **Changes in the manuscript** |
| | |

| 1.13 | **Reviewer comment** |
|---|---|
| | Some observations I am wondering whether the authors considered (and could thus address in the discussion):
 The use of XGBoost, rather than random forests, does limit the amount of variability in between trees. that is a pity as different trees can give different potential pathways to impact and thus account for different drought types. |
| | **Author's response** |
| | Thank you for pointing this out. We chose XGBoost because it is commonly considered a good match with SHAP, as indicated by the cited literature (Lundberg & Lee 2017, Yang et al., 2021; Jena et al., 2023; Raihan et al., 2023; Li et al., 2024). In a previous work (Brill et al., 2020), 5 different algorithms were used for modelling compound event damage, and results from Random Forest appeared quite difficult to interpret. In the experience of our first author, FB, the random subsets (bootstrapping) can lead to rather strange individual decision trees, especially when the sample size is small. The voting principle of Random Forest then leads to a good predictive model, according to the ensemble theory that many weak learners with uncorrelated errors tend to construct a strong learner. However, interpreting the individual weak learners might give misleading insights, in particular when the overall model skill is not very high.

 We added another sentence to support our choice of algorithm, and also added the concern of the reviewer to the discussion. |

| | **Changes in the manuscript** |
|---|---|
| | L263: "This iterative analysis of errors and weight-adjustment supposedly leads to models that reflect actual patterns in the overall data, rather than random patterns observed in random bootstrap subsets."

L563: "We chose the algorithm XGBoost, which, compared to Random Forest, limits the amount of variability between the individual decision trees. This is assumed to avoid erratic behavior, but on the other hand could also limit the potential damaging processes discovered by the models." |

| 1.14 | **Reviewer comment** |
|---|---|
| | The impact variable is continuous, rather that categorized or made boolean. that might have an influence on which types of nonlinearity the models can capture. |
| | **Author's response** |
| | Thank you. We added this point to the discussion |
| | **Changes in the manuscript** |
| | L560: "Both impact variables used in our regression are continuous rather than binary, which could affect the nonlinearities captured by the models." |

| 1.15 | **Reviewer comment** |
|---|---|
| | No accumulation times nor lag times were tested. This could also potentially improve the models to reflect diversity of drought types. |
| | **Author's response** |
| | As the investigated agricultural crops, as opposed to e.g. trees, are replaced every season, it does not seem logical to include a long accumulation time, except for:
    1.   The accumulation and time lag of soil drought in the deeper soil layers, which was accounted for by including the total soil drought magnitude. Figure 8 shows the lag of 1 year compared to SPEI
    2.   Seasonal SPEI was used in addition to monthly SPEI. However, the XGBoost models do not need the accumulated SPEI if the monthly layers are included, as the models can learn interactions that resemble accumulation.
The included combination of monthly SPEI, monthly top soil drought, and accumulated total soil drought indicators should provide the models a wealth of information from which to learn a diversity of drought types

We clarified this in the revised manuscript |
| | **Changes in the manuscript** |
| | L539: "From the monthly hazard features, the models can learn interactions that resemble accumulation – however, we did not include predictors from a previous year or even longer lag times. The only information on longer time is the SMI-Total (Fig. 8 shows the lag of 1 year compared to SPEI). As agricultural crops, as opposed to e.g. trees, are replaced every season, it does not seem logical to include longer lag times, but future research might investigate this." |

| 1.16 | **Reviewer comment** |
|---|---|
| | For crop losses in economic terms, were price shocks accounted for? |
| | **Author's response** |
| | Prices were only used for the monetary estimates, but the regression and SHAP analysis is based on relative yield gaps, i.e. without the prices. We included an additional sentence to clarify this |
| | **Changes in the manuscript** |
| | L557: "Regressions on county level are based on relative yield gaps. (…) Directly regressing economic loss would be possible, and lead to different insights (e.g. on the effect of price shocks)." |

| 1.17 | **Reviewer comment** |
|---|---|
| | The piece could end with some key take aways for farmers, for agricultural ministries and for drought disaster managers. now the suggestions are not very specific, but there are quite some learnings in the paper that could be translated into specific policy advises. |
| | **Author's response** |
| | Admittedly, this was a bit vague. Thank you pointing this out. We consider our study stronger on the technical side than on very practical recommendations, but we introduced a new paragraph on recommendations and also revised the ending of the Conclusion. |
| | **Changes in the manuscript** |
| | L574: "3.3.3 Recommendations
To prepare the agricultural sector, rural population and society for the uncertain future climate with an increased frequency of extreme hydrometeorological events, monitoring systems with early warnings are needed. Given that most decision makers, e.g. local authorities, disaster managers, or farmers, react to information about impacts (Dutt & Gonzales, 2010), such monitoring and early warning systems should be impact-based, rather than only inform about hazard. In particular we recommend to

1. Foster the implementation of impact-based monitoring and early warning systems for droughts to reduce impacts
2. Establish the use of interactive visualization tools in education and training to advance adaptation
3. Select drought-robust crops (farmers), e.g. rye over wheat; avoid adverse incentives (policy makers)
4. Provide water storage or other capacities for ad-hoc measures during the decisive summer months (here: June)"

L624: "Interactive visualization tools should enter the education system at all levels to train risk and climate literacy of future citizens, and demonstrate impacts of hazards rather than hazards only. Ultimately, interactive impact-based forecasting tools would offer a basis for science communication with policy makers and participatory modelling approaches to develop better climate policies and raise awareness for feasible adaptation options." |

| 2. | **Reviewer comment** |
|---|---|
| | In this study, the authors used a multi-index approach (exposure, risk, vulnerability) to model the impacts of drought events on agricultural systems in the German federal state of Brandenburg, considering the LST/NDVI ratio as the response variable. The scientific approach used is valid. It reflects the multifactorial complexity of the implications of drought for the productivity of the region's farming systems. |
| | **Author's response** |
| | Thank you for this positive overall evaluation.

For the sake of completeness, although the reviewer is certainly aware of this, we might add here that we even used 2 different response variables: the LST/NDVI ratio (per field) and relative yield gaps (per county). |
| | **Changes in the manuscript** |
| | |

| 2.01 | **Reviewer comment** |
|---|---|
| | « Empirical associations to the impact indicators on both spatial levels are compared. Non-linear models explain up to about 60% variance in the yield gap data, with lumped models for all crops being more stable than models for individual crops». This is imprecise, you must specify the names of the nonlinear models as well as for the grouped models. It is also important to include in the abstract the performance statistics of the models used. |
| | **Author's response** |
| | While we agree with the reviewer that model names and numbers can be important in many cases, we have to find a compromise here because the Abstract is limited to 200 words, and we would like the key messages in the abstract to be generally understandable by interested readers with less technical focus. In response to this request, we specified the model type from "non-linear" to "XGBoost" and added the precise score of the best model on county level "($R^2$ = 0.62)".
The general statement about the stability of the models refers to the figure 13 in the manuscript, where it is shown that distributions from repeated training with samples from all categories have a higher mean and lower variance than models repeatedly trained on thematic subsets such as individual crops or years. There is no single number to report, though, rather it is a condensed finding that we attempt to convey in language. We rephrased this part to make the findings clearer, and hope that the reviewer may agree with this attempt, or otherwise we are open for suggestions on how to further improve the abstract. |
| | **Changes in the manuscript** |
| | L21 now reads: "XGBoost models explain up to about 60% variance in the yield gap data (best $R^2$ = 0.62). Model performance is more stable for the drought years, and when using all crops for training rather than individual crops." |

| 2.02 | **Reviewer comment** |
|---|---|
| | "Rye is found less vulnerable than wheat, despite growing on poorer soils". The fact that rye grows on poorer soils is a proof that it is more resilient and less vulnerable than wheat, so I do not see why the conjunction of subordination although? |
| | **Author's response** |
| | This is a very good point. In fact, we find that rye on poor soils still is less affected by drought than wheat on good soil. It is one thing to assume one crop to be more or less vulnerable, or to find this in laboratory experiments, and another thing to substantiate this empirically under real world conditions. We added a sentence for clarification. |
| | **Changes in the manuscript** |
| | L25: "Rye is empirically found less vulnerable to drought than wheat, even on poorer soils."

L532: "While this already indicates that rye tolerates harsher conditions, we find empirically that rye on poor soil is still more robust under drought conditions in the region than wheat on good soil – based on both impact datasets." |

| 2.03 | **Reviewer comment** |
|---|---|
| | In introduction, « This has implications for modelling and Monitoring ». You mean implications in the modelling and monitoring of agricultural drought. If so, the sentence should be completed. |
| | **Author's response** |
| | Thank you for this observation. We completed the sentence accordingly |
| | **Changes in the manuscript** |
| | L103 The line now reads: "This has implications for modelling and monitoring of agricultural drought" |

| 2.04 | **Reviewer comment** |
|---|---|
| | Overall, the introduction is well written and argued. However, the application of artificial intelligence models in modelling drought impacts, risk, and vulnerability has been limited. It is worth adding a paragraph on the advantages and limitations of intelligence models in modelling the impacts of drought given that in your methodology you have used the extreme gradient boosting algorithm (XGBoost). |
| | **Author's response** |
| | Thank you. We revised the paragraph on methods in the introduction and included more discussion on advantages and limitations of AI models for drought impact prediction. Many of the references in the paragraph on methods use algorithms from the AI domain, especially Kondylatos et al. 2022, Peichl et al. 2021, Merz et al., 2013, Brill et al., 2020, Sodoge et al., 2023, Tanguy et al., 2023, among others.
Some parts in our Results & Discussion and Conclusion sections already addressed opportunities and limitations of AI methods, but in response to this reviewer request we made it more prominent in the revised manuscript. |

| | |
|---|---|
| | **Changes in the manuscript** |
| | L74: "from the field of (explainable) artificial intelligence (AI and XAI, respectively)" |
| | L82: "The application of AI methods in particular has led to considerable advances on the side on drought hazard monitoring and forecasting in recent years (Prodhan et al., 2022; Kowalski et al., 2023; Zhang et al., 2024). While these methods are very promising, they do rely on the availability of (big) data covering the processes of interest. On the side of vulnerability and impact-relevant factors, a key bottleneck of such data-driven studies is the availability of impact data." |
| | L563: "We chose the algorithm XGBoost, which, compared to Random Forest, limits the amount of variability between the individual decision trees. This is assumed to avoid erratic behavior, but on the other hand could also limit the potential damaging processes discovered by the models. For the models on county level, predictive features were derived by computing the relative area above/below evenly-spaced thresholds. An alternative here would be to use quantiles, or to automate the feature engineering by deep learning algorithms. Stronger AI methods, not only in the regression but also in the feature learning step (i.e. deep learning), could improve the predictive skill." |
| | L616 now reads: "Data-driven techniques from the AI domain can capture complex interactions in human-environments such as agriculture. SHAP plots uncover which factors drive the prediction of impact indicators in the models. This does not necessarily relate to causal effects in nature, though. We thus suggest to cross-check results obtained from different model setups, different regression targets, and ideally also different algorithms." |

| | |
|---|---|
| **2.05** | **Reviewer comment** |
| | *Line 250* «To retain as much information about the hazard distributions, we computed the relative affected area (non-)exceeding specified thresholds (in regular intervals of 0.5 for SPEI, 0.25 for LST/NDVI-anom., 0.05 for SMI, 5 for SMI-Total, and using the LBG class limits for AZL). A total of 68 features were created this way on county level». |
| | On what criterion were these thresholds considered? This deserves to be clarified. The different classification thresholds for these indices and their meanings should be provided in a table in the methodology section. |
| | **Author's response** |
| | True, this is a good point to further detail in the methods and also mention in the limitations of our study. The thresholds were chosen in regular intervals, which is a form of manual feature engineering. An alternative would be to use quantiles (less intuitive), or do automate feature engineering altogether via deep learning techniques. We added this in the discussion on limitations and included a table as requested. If the overall manuscript becomes too long, we could also shift this table to the Appendix. |
| | **Changes in the manuscript** |
| | Appendix B |

Table B. Intervals for thresholds

| Indicator category | Interval for thresholds (exact values) |
|---|---|
| SPEI | 0.5 (-4*., -3.5*, -3*, -2.5, -2, -1.5, -1, -0.5, 0) |
| SMI | 0.05 (0, 0.05, 0.10, 0.15) |
| SMI-Total | 5 (0, 5, 10, 15, 20, 25, 30, 35) |
| LST/NDVI-anom | 0.25 (0, 0.25, 0.50, 0.75, 1.00, 1.25, 1.50) |
| AZL | LBGs (23, 29, 36, 46) |

*only for SPEI-Magnitude

L565: "For the models on county level, predictive features were derived by computing the relative area above/below evenly-spaced thresholds. An alternative here would be to use quantiles, or to automate the feature engineering by deep learning algorithms."
* * *
| 2.06 | **Reviewer comment** |
|---|---|

The principle of the calculation of the LST/NDVI anomaly has not been sufficiently described. There should be a separate section to better describe and justify the choice of this anomaly to represent the impacts of drought when there are various other anomalies or indices that can better reflect the impacts of drought on agricultural systems. In this sense, the normalization indicated in Table 1 concerns only the LST values and/or the LST/NDVI values. If so, considering the max and min values or mean and standard deviation (SD)?

**Author's response**

It is certainly true that other indicators could be used for observing drought impacts, and we pointed this out once more in the revised manuscript by adding additional references on alternative metrics.

However, we are convinced that the provided literature references contain sufficient justification for using LST/NDVI. L91 in the preprint: "The ratio between LST and NDVI is a well-established observable indicator for that purpose (McVicar and Bierwirth, 2001; Karnieli et al., 2010; Crocetti et al., 2020). Mid growing season is generally regarded as the most decisive time of observation (Ghazaryan et al., 2020)."

In line with the reviewer comment, one of the conclusions of our paper is that better remote sensing indicators should be explored. L610 in the preprint: "Monitoring and impact-based forecasting are needed to prepare for future hazards, which can hardly be mitigated. Stronger remote sensing indicators on drought impacts seem necessary in that context."

The anomaly was calculated as difference between the field-level values and the area-weighted mean for a specific crop, so that values for different crops can be better compared. We included a formula in the manuscript as requested. In addition, we would like to mention once more that the exact procedure of our study is documented in the provided programming scripts, which are publicly accessible via Github

We would also like to highlight that we used 2 different impact indicators and compared them – the other one being based on reported yields. The strongest point of our study might be this

comparison, which implicitly addresses potential shortcomings of individual impact indicators and insights derived thereof.

| Changes in the manuscript |
|---|
| Introduction: "While there are various potential indicators for mapping drought impacts on crops, the ratio between LST and NDVI is a particularly well-established observable metric for that purpose (McVicar and Bierwirth, 2001; Karnieli et al., 2010; Crocetti et al., 2020)."

New subsections at L194: "2.4.1 LST/NDVI Anomaly", and L212: "2.4.2 Empirical yield gaps"

L206:
"

$$LST/NDVI_{anom,f,y} = \frac{LST/NDVI_{c,f,y} - \overline{LST/NDVI_c}}{\overline{LST/NDVI_c}} \quad (Eq.2)$$
Where $I_{anom,\ f,\ y}$ is the $\overline{I_c}$ is the area-weighted mean for a given crop across all years, and the subscripts c, f, and y denote crop, field, and year, respectively.

"

L554: "The choice of impact variables, and preprocessing thereof, might introduce biases. LST/NDVI anomaly is a commonly used indicator for drought-related crop health, but others are possible, such as the radar vegetation index (Kim et al., 2012), hyperspectral metrics (Dao et al., 2021), fractional cover time series (Kowalski et al., 2023), or multimodal techniques (Karmakar et al., 2024)."

L615: "Stronger remote sensing indicators on drought impacts, beyond LST/NDVI, seem necessary as well" |

| 2.07 | Reviewer comment |
|---|---|
| | Ligne 255-260 « In 2013 and 2014 the SMI-Total is close to 0, observed vegetation health is at its maximum (i.e. negative LST/NDVI-anom.), essentially no impact-related statements…..» Similarly, to better assess the consistency of these statements, the formula and principle of the calculation of the IMS and IMS-Total must be clearly described in the methodological section with the different classification thresholds. |
| | **Author's response** |
| | Thank you for pointing this out. We recognize that our description of SMI-based features was not as clear as it should be. We added the formula for the soil drought intensity and a description. More details of the calculation are described in the referenced article by Boeing et al. (2022).

It might be slightly confusing that we use the SPEI absolute values, and the SMI-derived drought intensity, as well as the SMI-derived drought magnitude for the total soil, but still refer to the values as "SMI" rather than "SMI-based drought intensity". We are open for renaming the features (for example to "SDI" or "SMDI"), if the reviewers consider this more understandable.

Please note that normalization does not have any effect on the XGBoost models, as they operate with relative differences rather than absolute values. |
| | **Changes in the manuscript** |

L182 The paragraph now reads: "Identical to the SPEI data, we use monthly values and a growing season aggregation of drought intensity derived from the soil moisture index (SMI) for the top soil (25 cm), again from March to July. To add some information on slower long-term drought processes (i.e. accumulation and lag time), we further include the annual drought magnitude for the total soil (up to 1.8 m depth), which is temporally aggregated from April to October (SMI-Total). SMI drought intensity represents the integrated area below the 20th percentile of the soil moisture index for a given time (and area). The general formula, as presented in Boeing et al. (2022), includes a potential normalization over the area of investigation (Eq. 1). In contrast to the drought intensity, the drought magnitude is not normalized (i.e. shift of absolute values but same relative order). However, in this study we used the individual raster cell values, which implies that no area normalization is performed either way.

$$SMI = \frac{1}{d \cdot A} \sum_{t_0}^{t_1} \int_A [\tau - SMI^*_i(t)]_+$$

where τ is the drought threshold, SMI* is the raw soil moisture index, and d and A refer to the duration and area of potential aggregation, respectively. A value of 0 for all SMI-based features thus means, that none of the values were below drought threshold τ. We use τ = 0.2 (20th percentile), which is a common value for drought analysis adopted in the literature (e.g. US drought monitor, Svoboda et al, 2002). For more details, the interested reader is referred to Boeing et al. (2022)."

| 2.08 | **Reviewer comment** |
|---|---|
| | In Table 1, you mentioned that the monthly SPEI used has a resolution of 10 km and the source is the reference Zhang et al. (2024). However, in this reference, the SPEI used has a 1 km resolution. It is a bit ambiguous. Has the SPEI been calculated? or was the same database from the Zhang et al. (2024) study used? If this is the case, the spatial resolution of 10 km should be rectified because in the source reference mentioned it is rather 1 km that is mentioned. |
| | **Author's response** |
| | Thank you for this observation. We apologize for the confusion. We used the same method and code to recalculate SPEI as in Zhang et al. (2024), including data transformation and quality check, but the data are actually on 10 km, based on the 0.1° spatial resolution E-OBS dataset by Cornes et al. (2018). We revised Line 165 in the main manuscript to make it clear for readers. We added the proper reference (Cornes et al., 2018) in "Data source and references" in Table 1 and the Reference section.

Author HZ found the accuracy of this SPEI dataset to be higher than the one used in her previous publication (Zhang et al., 2024), and now also uses this in another ongoing study (Zhang et al., in review). The data can be viewed in the provided R-Shiny app. |
| | **Changes in the manuscript** |
| | L173 "Monthly values of SPEI-1 (one-month accumulation SPEI) used in this study are at a 10 km grid resolution from 2013 to 2022 based on the E-OBS dataset (Cornes et al., 2018). The calculation detail can be referenced in Zhang et al. (2024)."

Table 1: "(Cornes et al. 2018)"

References: "Cornes, R. C., et al. 2018. An ensemble version of the E-OBS temperature and 239 precipitation data sets. Journal of Geophysical Research: Atmospheres, 123(17), 9391-240 9409." |

| 2.09 | **Reviewer comment** |
|---|---|
| | The algorithm used to calculate the Landsat LST was not explained in the methodology |
| | **Author's response** |
| | The data was taken from Google Earth Engine Landsat 8 L1T2. We included a reference to the procedure (Cook et al. 2014), although the data description from the USGS might be more exhaustive. The reference is in the Table 1. The respective code in online on GitHub. |
| | In response to this reviewer comment, we made the data reference more obvious in the manuscript. |
| | Data: https://developers.google.com/earth-engine/datasets/catalog/LANDSAT_LC08_C02_T1_L2#description |
| | Code: https://github.com/fabiobrill/brandenburg-drought-study/blob/main/preprocessing/gee_landsat8_dlschema.js |
| | **Changes in the manuscript** |
| | L196: "This dataset already includes processed LST (Cook et al., 2014)." |
| | Table 1: "https://developers.google.com/earth-engine/datasets/catalog/LANDSAT_LC08_C02_T1_L2#description" |

**Authors Response to the Editor:**

| 3. | **Editor comment** |
|---|---|
| | The research presented in this study is new and innovative. The methods are adequate and the results convincing. The material is well presented. Both reviewers and the editor support publication after minor revisions. The changes to the manuscript suggested by the authors as a response to the reviews are conclusive. |
| | **Author's response** |
| | Thank you very much for your positive evaluation of our revised manuscript. We appreciate your efforts that helped to improve the earlier version. |
| | **Changes in the manuscript** |
| | |

| 3.1 | **Editor comment** |
|---|---|
| | Please revise the manuscript as proposed in your responses to the reviews. I suggest to include table 2 (Intervals for thresholds) in the appendix. |
| | **Author's response** |
| | We have revised the manuscript accordingly. We have also followed your suggestion and moved table 2 to the appendix. |

| | **Changes in the manuscript** |
|---|---|
| | Table 2 is now Appendix B, |
| | former Appendix B is now Appendix C, |
| | former Appendix C is now Appendix D |

| | |
|---|---|
| **3.2** | **Editor comment** |
| | I have one additional remark for the revised version. Please give a reference for the following statement or rephrase it: |
| | "... project a shift in seasonality of rainfall: more in winter, and less during summer months." |
| | This statement is for example not supported by Jacob et al. 2014 (supplementary material of DOI 10.1007/s10113-013-0499-2) |
| | It might be that the problem is probably not so much the mean amount of precipitation but rather the shift to more extreme events during summer where the water is not able to infiltrate the soil. |
| | **Author's response** |
| | We very much appreciate this observation, as it allows us to sharpen the context information to one thematic key point! |
| | Indeed the dynamical RCMs (regional climate models) from Eurocordex (dynamically downscaled CMIP-5 global climate models (GCM)) as shown in Jacob et al (2014) generally show large uncertainty for changes in summer precipitation. The forcing GCMs partly show stronger summer drying than Eurocordex RCMs. The summer drying is a bit stronger even for GCMs from CMIP-6 than CMIP-5. See Eurocordex, CMIP-5 and CMIP6 comparison for Europe in Coppola et al. (2021) |
| | Another reason for the impression of strong future summer drying in Brandenburg may result from that fact that many studies conducted in the past for Germany used Statistical Models for downscaling the GCMs (mainly Stars and Wettreg) that generally project much stronger drying. These model results are for example widely used by government institutions. There is a project report from Reklies-DE showing the differences between dynamical and statistical model results: https://reklies.hlnug.de/fileadmin/tmpl/reklies/dokumente/ReKliEs-De-Ergebnisbericht.pdf |
| | Table 2-2 on page 11 of the official climate adaptation strategy paper by Brandenburg's MLUK contains ensemble-derived mean values for mid-century and end-of-century, in which summer precipitation (German: "Sommerniederschlag") is estimated with +0% (i.e. no change) and +1%, respectively, but the climatic water balance in summer (German: "Klimatische Wasserbilanz Sommer") is estimated at -12 mm and -22 mm, respectively: https://mluk.brandenburg.de/sixcms/media.php/9/Klimaanpassungsstrategie-Brandenburg-LF.pdf |
| | On top of this changing climatic water balance comes the issue of infiltration, as mentioned by the Editor. Author F.B(2) is currently preparing a manuscript on climate change related SM drought changes in Germany using a large set of RCMs, however the study is not published yet. Studies by Berg et al. (2017) and Cook et al. (2018) show differences between precipitation and soil moisture changes under climate change, based on CMIP-5 and CMIP-6, respectively. In particular, Cook et al. (2018) write that the modelled drying of the top soil is more robust and widespread than modelled changes in precipitation. |
| | We rephrased the sentence and included more references. |
| | **Changes in the manuscript** |

L518: The paragraph now reads:

"This is of particular concern as current regional climate simulations for Brandenburg project a shift in seasonal water balance and intensity of rainfall: more precipitation than today might arrive in winter, but rainfall during the summer months is expected to occur in shorter and more intense downpours, which implies a lower fraction of infiltration and longer times of amplified evaporative loss between the rain events (Jacob et al., 2014; Coppola et al., 2021; MLUK, 2023). While uncertainties of these projections are rather high for the case of summer precipitation, more robust projections of increasing summer dryness have been shown for surface soil moisture (Berg et al., 2017; Cook et al., 2018).

We included these additional references in the revised manuscript:

Berg, A., Sheffield, J., and Milly, P. C. D.: Divergent surface and total soil moisture projections under global warming. Geophys. Res. Lett., 44 (1), 236–244, https://doi.org/10.1002/2016gl071921, 2017.

Cook, B.I., Mankin, J.S., and Anchukaitis, K.J.: Climate change and drought: From past to future. Current Climate Change Reports, 4(2), 164–179, https://doi.org/10.1007/s40641-018-0093-2, 2018.

Coppola, E., Nogherotto, R., Ciarlo', J. M., Giorgi, F., van Meijgaard, E., Kadygrov, N., et al.: Assessment of the European Climate Projections as Simulated by the Large EURO-CORDEX Regional and Global Climate Model Ensemble. JGR: Atmospheres, 126(4), e2019JD032356. https://doi.org/10.1029/2019JD032356, 2021.

Jacob, D., Petersen, J., Eggert, B. et al.: EURO-CORDEX: new high-resolution climate change projections for European impact research. Reg. Environ. Change, 14, 563–578, https://doi.org/10.1007/s10113-013-0499-2, 2014.

| 4. | **Review file validation** |
|---|---|
| | Please ensure that the colour schemes used in your maps and charts allow readers with colour vision deficiencies to correctly interpret your findings. Please check your figures using the Coblis – Color Blindness Simulator (https://www.color-blindness.com/coblis-color-blindness-simulator/) and revise the colour schemes accordingly. |
| | **Author's response** |
| | We have checked our figures with the suggested Coblis tool and assume that the colors used can be distinguished by most people – however, this tool simulates 8 different types of colorblindness and does not provide a simple evaluation like "correct" or "incorrect". If the production team of the journal requests color changes to a particular figure of our manuscript, we promise to provide these changes. |
| | **Changes in the manuscript** |
| | |